# Diffusion-based Decoupled Deterministic and Uncertain Framework for Probabilistic Multivariate Time Series Forecasting

**Qi Li** [1,*] **Zhenyu Zhang** [1,*] **Lei Yao**[1], **Zhaoxia Li**[3], **Tianyi Zhong**[4], **Yong Zhang**[1,2 †]
[1]Beijing University of Posts and Telecommunications
[2]Beijing Key Laboratory of Work Safety Intelligent Monitoring
[3]China Unicom Cloud Data Co., Ltd., [4]University of Melbourne
{li.q,zhangzhenyucad,Yao_Lei,yongzhang}@bupt.edu.cn
lizx101@chinaunicom.cn, zhong17@student.unimelb.edu.au

## Abstract

Diffusion-based denoising models have demonstrated impressive performance in probabilistic forecasting for multivariate time series (MTS). Nonetheless, existing approaches often model the entire data distribution, neglecting the variability in uncertainty across different components of the time series. This paper introduces a **D**iffusion-based **D**ecoupled **D**eterministic and **U**ncertain ($D^3U$) framework for probabilistic MTS forecasting. The framework integrates non-probabilistic forecasting with conditional diffusion generation, enabling both accurate point predictions and probabilistic forecasting. $D^3U$ utilizes a point forecasting model to non-probabilistically model high-certainty components in the time series, generating embedded representations that are conditionally injected into a diffusion model. To better model high-uncertainty components, a patch-based denoising network (PatchDN) is designed in the conditional diffusion model. Designed as a plug-and-play framework, $D^3U$ can be seamlessly integrated into existing point forecasting models to provide probabilistic forecasting capabilities. It can also be applied to other conditional diffusion methods that incorporate point forecasting models. Experiments on six real-world datasets demonstrate that our method achieves over a 20% improvement in both point and probabilistic forecasting performance in MTS long-term forecasting compared to state-of-the-art (SOTA) probabilistic forecasting methods. Additionally, extensive ablation studies further validate the effectiveness of the $D^3U$ framework.

## 1 Introduction

Multivariate time series (MTS) are prevalent in real-world applications. Probabilistic MTS forecasting is widely used as a decision support in domains such as finance (Wiese et al., 2020), healthcare (Teng et al., 2020) and power energy (Nowotarski & Weron, 2018). Recently, autoregressive models (Rasul et al., 2021; Li et al., 2022; Tashiro et al., 2021) have achieved significant success in short-term forecasting tasks within the field of deep learning-based probabilistic time series forecasting. Nonetheless, the low generation efficiency of autoregressive models over long sequences limits their application in long-term forecasting tasks. To address this challenge, non-autoregressive approaches have started to gain attention. Shen & Kwok (2023) proposes a non-autoregressive diffusion model, introducing two novel conditioning mechanisms to achieve high-quality long-term time series forecasting. Li et al. (2024a) designs a non-autoregressive framework that integrates a conditional diffusion process with a Transformer model to enable distributional forecasting for MTS data. Shen et al. (2024) decomposes time series into multiple scales, incorporating them into both the forward and reverse processes of the diffusion model, enabling probabilistic forecasting through non-autoregressive denoising.

---

*Equal contribution. Code will be open-sourced at https://github.com/zhangzhenBrave/D3U.
†Correspondence to Yong Zhang <yongzhang@bupt.edu.cn>.

However, the aforementioned methods pay less attention to the differences in uncertainty across various components of time series data. We provide two case studies to analyze these differences:

(1) The first case study focuses on two representative point forecasting models (Liu et al., 2022; Nie et al., 2023). The input series is first decomposed into trend, seasonal, and residual components, denoted as $X_T$, $X_S$, and $X_r$, respectively (see Appendix B.1 for details on the decomposition method). Each model is then trained using both the original series $X$ and the series with the residual component removed, $X_T + X_S$, while keeping all other model parameters and settings unchanged. The model performances under both conditions are compared in Fig. 1a.

(2) In the second case study, a patch-based vector quantization (VQ) model (Zhao et al., 2024) is employed to further analyze the uncertainty of the residual component. Specifically, the VQ model is first trained using $X_T + X_S$. Subsequently, several patches are randomly selected from the training set and input into the trained VQ model in two forms: without decomposition ($X$) and with decomposition ($X_T + X_S$), to observe their quantization distributions. The Kullback-Leibler (KL) divergence between the quantization distribution and a uniform distribution is used as a measure of data uncertainty. A larger KL divergence indicates that the input sequence is better quantized and more likely to exhibit lower uncertainty. Several examples are illustrated in Fig. 1b

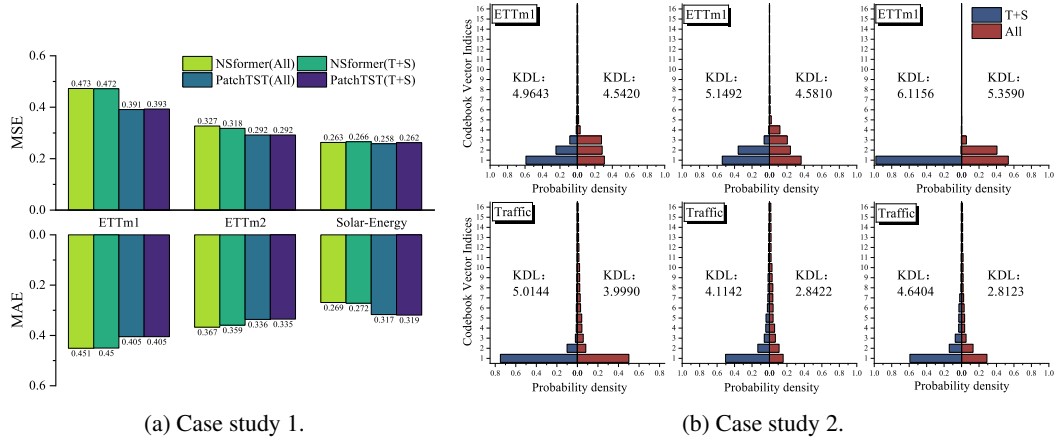

(a) Case study 1.  (b) Case study 2.

Figure 1: Differences in modeling difficulty and uncertainty across different components of time series. (a) Both models show no significant changes in predictive performance after the removal of residual components. 'All' indicates training with the original series, while 'T+S' indicates training without the residual component. Results are averaged from all prediction lengths {96,192,336,720}. (b) Quantization distribution comparison of time series patches. KDL denotes the KL divergence. 'All' means containing residuals, 'T+S' means no residuals.

The results of the above case studies provide two key insights:

- From the perspective of series decomposition, point forecasting models exhibit varying capabilities in modeling $X_T$, $X_S$, and $X_r$. When only the trend and seasonal components are retained as input, the performance of these models does not significantly degrade and, in some cases, even improves. This suggests that the models are less effective at modeling the residual component compared to the trend and seasonal components.

- The difficulty in modeling the residual component appears to be related to the uncertainty it contains. Experiments using VQ-based point forecasting models show that the residual information increases the complexity of quantizing the input series. By randomly selecting five batches of data from the ETTm1 and Traffic datasets, the quantization distribution of all patches within each batch is analyzed. Results reveal that over 57% and 56% of the patches in the two datasets, respectively, exhibit worse quantization performance after the residual component is reintroduced. This suggests that the residual component of time series data tends to contain more uncertainty than the trend and seasonal components.

Based on insights from the aforementioned case studies, we propose a **D**iffusion-based **D**ecoupled **D**eterministic and **U**ncertain (D³U) framework for long-term MTS probabilistic forecasting. The

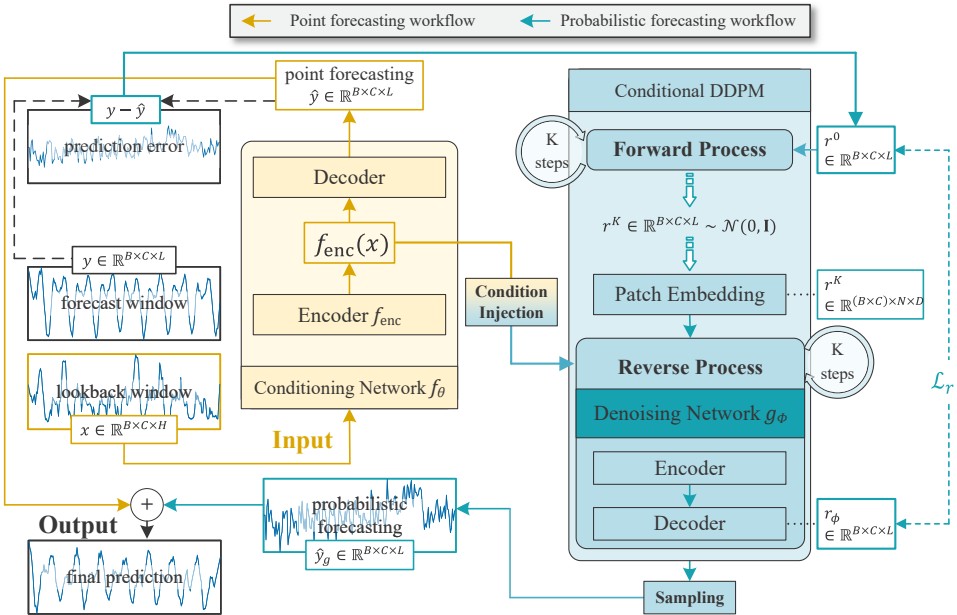

Figure 2: An illustration of the proposed $D^3U$ framework. The framework consists of two main components: the conditioning network and the conditional Denoising Diffusion Probabilistic Model (DDPM). The conditioning network is responsible for modeling the high-certainty components of the data, providing condition information for the reverse process of the conditional DDPM. The conditional DDPM focuses on modeling the distribution of the high-uncertainty components.

$D^3U$ framework leverages pre-trained point forecasting models for non-probabilistic modeling of high-certainty components in the data, while employing a diffusion-based generative model to capture the probabilistic distribution of high-uncertainty components. The $D^3U$ framework offers a key advantage: The pre-trained point forecasting model provides strong point forecasting capabilities for the overall framework and generates useful representations of high-certainty components, which serve as conditional information for the diffusion denoising model. This enhances the diffusion model's ability to capture the data distribution. Note that Li et al. (2024a)(TMDM) also combines a pre-trained point prediction model with a Transformer-based diffusion model. The most significant difference between $D^3U$ and TMDM lies in their approach to modeling uncertainty in time series. TMDM directly models the overall uncertainty in the data, without focusing on the differences in uncertainty across different components. In contrast, $D^3U$ adopts a divide-and-conquer approach to components with varying levels of uncertainty in the time series. In $D^3U$, the diffusion model is only used to model the data distribution of high-uncertainty components, simplifying the difficulty of modeling the distribution and enables stable forecasting with fewer sampling steps. Additionally, the pre-trained model can be frozen during training, thereby reducing the overall computational cost.

Our contributions are summarized as follows:

1. Motivated by the phenomenon observed in case studies that point forecasting models exhibit varying capabilities in modeling the deterministic and uncertain components of time series, we propose a novel complementary modeling approach that combines point forecasting models and probabilistic forecasting models from the perspective of decoupling the deterministic and uncertain components of time series data. Specifically, point forecasting models and probabilistic forecasting models are used to model the high-certainty and high-uncertainty components of time series data, respectively.

2. We propose $D^3U$, a long-term MTS probabilistic forecasting framework based on the complementary modeling. $D^3U$ leverages a pre-trained point forecasting model to learn the high-certainty components and injects valuable representations of these components as

conditional information into the reverse process of a conditional DDPM, aiding it in performing probabilistic forecasting on the high-uncertainty components. The final prediction combines the probabilistic forecast with the non-probabilistic forecast from the point forecasting model, enabling $D^3U$ to achieve accurate data distribution modeling while maintaining strong point forecasting performance. $D^3U$ is a plug-and-play framework that can be seamlessly integrated with existing point forecasting models and diffusion-based long-term forecasting models, improving both point and probabilistic forecasting capabilities.

3. Within the $D^3U$ framework, we design a patch-based denoising network, PatchDN, to enhance the diffusion model's ability to represent the high-uncertainty components in time series data. Our method demonstrates excellent probabilistic forecasting performance and competitive point forecasting capabilities across six real-world datasets.

## 2 BACKGROUND

### 2.1 UNCERTAINTY IN POINT FORECASTING MODELS

Point forecasting models are non-probabilistic models in nature. Given a time series input $X$, these models directly learn a mapping from the input space to the output space, providing the conditional expectation of the prediction target, $\mathbb{E}(\hat{Y}|X) := F(X)$, where $F(\cdot)$ is a parameterized function and $\hat{Y}$ is the model's output. We analyze the input series from the perspective of time series decomposition (Petropoulos et al., 2022), decomposing it as follows: $X = X_{nf} + \epsilon_X$, where $\epsilon_X$ represents the inherent noise in $X$, and $X_{nf}$ is the ideal noise-free time series. Further decomposition of $X_{nf}$ yields $X_T + X_S$, where $X_T$ and $X_S$ represent the trend and seasonal components, respectively. In an ideal scenario, the inherent noise $\epsilon_X$ corresponds to the residual component, $X_R$. However, in practice, it is not possible to fully separate $\epsilon_X$ from $X$, nor to obtain purely trend and seasonal components. Therefore, we decompose the residual as follows: $X_R = X_r + \epsilon'_X$, where $X_r$ represents the non-noise component of the series after removing the evident trend and seasonal information, and $\epsilon'_X$ is the noise component of $\epsilon_X$, excluding the noise that remains in $X_T$ and $X_S$.

The prediction error of a point forecasting model, $\|Y - \hat{Y}\|$, includes both the error caused by insufficient modeling of $X_T$, $X_S$, and $X_r$, as well as the error due to the inherent noise $\epsilon'_X$. Therefore, $\|Y - \hat{Y}\|$ can be viewed as a combination of aleatoric uncertainty and epistemic uncertainty (Li et al., 2022), encompassing a majority of the uncertainty information in the sequence that is difficult for point forecasting models to capture.

### 2.2 PATCH IN TIME SERIES FORECASTING MODELS

In recent years, advancements in natural language processing (NLP) (Radford et al., 2019) and computer vision (CV) (Dosovitskiy et al., 2020) have inspired the development of many patch-based time series forecasting models. Nie et al. (2023); Zhang & Yan (2023) segment input time series into subseries-level patches and encode them as embedded representations for long-term forecasting tasks. Zhou et al. (2023) further employs GPT-2 as a tokenizer to generate discrete embedded representations of patches. Jin et al. (2023); Chang et al. (2023) propose methods to align time series patches with pre-trained large language models (LLMs), improving the discrete representations of time series. The use of patches for long-term forecasting has gained widespread consensus, as patches are more effective at extracting local semantic information. However, current diffusion-based probabilistic forecasting models for time series (Shen & Kwok, 2023; Li et al., 2024a) primarily focus on pointwise modeling of time series data in the design of the denoising network.

### 2.3 DIFFUSION-BASED PROBABILISTIC TIME SERIES FORECASTING

Diffusion-based probabilistic models (Sohl-Dickstein et al., 2015) have recently gained prominence as a potent method in generative modeling. The DDPM (Ho et al., 2020), a widely known diffusion model, is extensively applied in probabilistic generation tasks. DDPM consists of two primary processes: the forward (diffusion) process and the reverse (denoising) process. Given an input vector $y_0$, the forward process gradually adds zero-mean Gaussian noise into $y_0$ following a Markov

chain over $K$ steps:

$$q(y^{1:K}|y_0) := \prod_{k=1}^{K} q(y^k|y^{k-1}), \ q(y^k|y^{k-1}) := \mathcal{N}(y^k; \sqrt{1-\beta_k}\, y^{k-1}, \beta_k \mathbf{I}), \ k = 1, \ldots, K, \quad (1)$$

where $\beta_k$ is a small positive constant denoting the variance of the Gaussian noise added at step $k$.

In practice, $y^k$ can be sampled directly from $y_0$ in a single step:

$$q(y^k|y_0) = \mathcal{N}(y^k; \sqrt{\bar{\alpha}_k}\, y_0, (1-\bar{\alpha}_k)\mathbf{I}), \ y^k = \sqrt{\bar{\alpha}_k}\, y_0 + \sqrt{1-\bar{\alpha}_k}\, \epsilon, \quad (2)$$

where $\bar{\alpha}_k := \prod_{i=1}^{k} \alpha_i$ with $\alpha_i := 1 - \beta_i$, and $\epsilon$ is noise sampled from $\mathcal{N}(0, \mathbf{I})$.

The reverse process aims to recover the original data $y_0$ from the noisy sample $y^k$ through a denoising procedure. This process is formulated as a Markov chain with learned transitions, where the noise is progressively removed at each step:

$$p_\phi(y^{0:K}) := p(y^K) \prod_{k=1}^{K} p_\phi(y^{k-1}|y^k), \ p_\phi(y^{k-1}|y^k) := \mathcal{N}(y^{k-1}; \mu_\phi(y^k, k), \Sigma_\phi(y^k, k)) \quad (3)$$

The variance $\Sigma_\phi(y^k, k)$ is typically fixed as $\sigma_k^2 \mathbf{I}$, while the mean $\mu_\phi(y^k, k)$ is parameterized by a neural network with parameters $\phi$. This network is generally used for noise estimation or data prediction. Once the model is trained, samples can be iteratively drawn from the reverse process $p_\phi(y^{k-1}|y^k)$ to reconstruct $y_0$.

## 3 PROPOSED METHOD

In this section, we present $D^3U$, a novel diffusion-based probabilistic MTS forecasting framework that decouples the deterministic and uncertain components of the MTS data. As illustrated in Fig. 2, $D^3U$ consists of two main parts: a pre-trained point forecasting model (conditioning network) and a patch-based conditional DDPM.

Given a history MTS data $x_{1:H} \in \mathbb{R}^{C \times H}$ and the corresponding prediction target $y_{1:L} \in \mathbb{R}^{C \times L}$, where $H$ and $L$ denote the lookback window size and forecast window size, respectively. $C$ denotes the number of variables in the MTS data. We use the conditioning network, denoted as $f_\theta$, to obtain the conditional expectation $\hat{y} := \mathbb{E}(\hat{y}|x_h; \theta) \in \mathbb{R}^{C \times L}$ for the high-certainty components, where $x_h$ can either be the $x_T + x_S$ or directly the original series $x_{1:H}$. The specific choice depends on the modeling preferences and capabilities of the point forecasting model. We assume that the prediction error of $f_\theta$, $\|y - \hat{y}\|$, contains most of the high-uncertainty components that are difficult to model using a point forecasting model. The conditional DDPM is used to model these components' distribution $p_\phi(y - \hat{y}|f_{\text{enc}}(x_h); \phi)$, where $\phi$ denotes the parameters of the denoising network $g_\phi$. Here, $f_{\text{enc}}$ denotes the encoder of $f_\theta$ and $p_\phi(y - \hat{y}|f_{\text{enc}}(x_h))$ is the conditional probability density function of $y - \hat{y}$. $f_{\text{enc}}$ injects information from $x_h$ as a condition into $g_\phi$. The final prediction is $\hat{y}_g + \hat{y}$, where $\hat{y}_g$ is sampled from $p_\phi$.

### 3.1 CONDITIONING NETWORK

In the $D^3U$ framework, one of the primary roles of the conditioning network is to extract useful information from $x_h$ to effectively guide the denoising process in the conditional DDPM. Drawing inspiration from advancements in image generation (Rombach et al., 2022; Radford et al., 2021), the conditioning network is implemented as a pre-trained, well-established point forecasting model, such as NSformer (Liu et al., 2022) or PatchTST (Nie et al., 2023). Moreover, existing MTS conditional DDPM models (Shen & Kwok, 2023; Li et al., 2024a) typically utilize $\hat{y}$ as the conditional input for the denoising network. However, this approach may overlook valuable intermediate features, particularly in long-term forecasting tasks. Experimental results demonstrate that leveraging $f_{\text{enc}}(x_h)$ as the guiding signal provides more effective guidance, improving the denoising process and enhancing prediction accuracy.

In the following content, we consistently use SparseVQ (Zhao et al., 2024) as the conditioning network. SparseVQ is a Transformer-based time series model that integrates VQ (Van Den Oord et al., 2017) with patching techniques, demonstrating strong predictive performance on MTS data.

### 3.2 CONDITIONAL DDPM AND PATCH DENOISING NETWORK

#### 3.2.1 CONDITIONAL DDPM FOR RESIDUAL DISTRIBUTION MODELING

In $D^3U$, the conditional DDPM is employed to model the distribution of residual components in the prediction target. In practice, the prediction error of the conditioning network is used as the residual:

$$r_{1:L}^0 := y - \hat{y} = y - f_\theta(x_h) \tag{4}$$

Given the input residual components $r_{1:L}^0 \in \mathbb{R}^{C \times L}$ and the condition $c = f_{\text{enc}}(x_h)$, the conditional DDPM performs residual prediction by modeling the following distribution:

$$p_\phi\left(r_{1:L}^{0:K} \mid c\right) = p_\phi\left(r_{1:L}^K\right) \prod_{k=1}^{K} p_\phi\left(r_{1:L}^{k-1} \mid r_{1:L}^k, c\right) \tag{5}$$

The reverse process at step $k$ is defined as:

$$p_\phi(r_{1:L}^{k-1} \mid r_{1:L}^k, c) = \mathcal{N}(r_{1:L}^{k-1}; \mu_\phi(r_{1:L}^k, k \mid c), \sigma_k^2 \mathbf{I}), \tag{6}$$

The mean $\mu_\phi(r_{1:L}^k, k \mid c)$ is parameterized as:

$$\mu_\phi(r_{1:L}^k, k \mid c) = \frac{\sqrt{\alpha_k}(1 - \bar{\alpha}_{k-1})}{1 - \bar{\alpha}_k} r_{1:L}^k + \frac{\sqrt{\bar{\alpha}_{k-1}}\beta_k}{1 - \bar{\alpha}_k} r_\phi(r_{1:L}^k, k \mid c), \tag{7}$$

where $r_\phi$ is predicted by a patch-based denoising network $g_\phi$ that models the distribution of $r_{1:L}^0$. The learnable parameters $\phi$ are optimized by minimizing the following loss function:

$$\mathcal{L}_r = \mathbb{E}_{r_{1:L}^0, \epsilon, k, c} \left[ \| r_{1:L}^0 - r_\phi(r_{1:L}^k, k \mid c) \|^2 \right]. \tag{8}$$

Next, we present the design of an effective denoising network $g_\phi$.

#### 3.2.2 PATCH-BASED DENOISING NETWORK

To better represent MTS data, we design a patch-based denoising network (PatchDN). PatchDN adopts a simple channel-independent (CI) setting (Nie et al., 2023; Woo et al., 2024; Goswami et al., 2024).

**Patching :** Specifically, the residual data $r_{1:L}^0$ is firstly divided into equilong patches. Let $P$ represent the patch length, and $S$ the stride, i.e., the non-overlapping portion between consecutive segments. The generated sequence is denoted as $r_p^k \in \mathbb{R}^{C \times N \times P}$, where $C$ and $N$ represent the number of variables and patches, respectively, $N = \left\lfloor \frac{(H-P)}{S} \right\rfloor + 2$ (Nie et al., 2023). Next, these patches will undergo the Patch Embedding: patches are projected into the latent space through a linear layer $W_p \in \mathbb{R}^{P \times D}$, and fixed transformer's sinusoidal positional embeddings $W_{pos} \in \mathbb{R}^{1 \times N \times D}$ (Vaswani, 2017) are added to incorporate positional information. After Patch Embedding, the representation can be expressed as:

$$r_{\text{PE}}^k = \text{Patch\_Embedding}(r_{1:L}^k) \in \mathbb{R}^{C \times N \times D}. \tag{9}$$

**Time Embedding:** As in (Shen & Kwok, 2023; Li et al., 2024a), the representation of the diffusion steps is obtained using the transformer's sinusoidal position embedding (Vaswani, 2017), followed by two fully connected layers to project it into the latent space.

**Condition Injection:** In the reverse process, effectively incorporating conditional information can guide the model to gradually denoise and generate time series samples that align with the given conditions. The patch-based denoising network adopts a CI strategy, making the recovery of different scales across variables particularly important. We aim to leverage conditional information to apply fine-grained control over the scaling and shifting of different dimensions, thereby improving the recovery of noise-free time series samples.

In the field of CV, FiLM (Feature-wise Linear Modulation) (Perez et al., 2018) is an effective technique that incorporates conditional information by dynamically adjusting the mean and variance in normalization layers, such as batch normalization and layer normalization. Similarly, the DiT model (Peebles & Xie, 2023) has demonstrated the effectiveness of this approach in generating high-quality image samples. Inspired by this design, we also introduce conditional information by controlling the

scale and shift parameters of the adaptive layer normalization (AdaLN) layers in the Transformer encoder. Unlike the DiT model, which adjusts based on the batch dimension, the AdaLN layers in PatchDN dynamically adjust each variable dimension to handle the non-stationarity and heterogeneity commonly found in time series data. The process can be described as follows:

$$h = p^k + \text{Reshape}(c) \in \mathbb{R}^{C \times 1 \times D}. \quad c_{scale}, c_{shift} = \text{Linear}(h) \in \mathbb{R}^{C \times 1 \times D}, \tag{10}$$

$$\text{AdaLN}(r_{\text{PE}}^k, h) = c_{scale}\text{LayerNorm}(r_{\text{PE}}^k) + c_{shift} \in \mathbb{R}^{C \times N \times D}, \tag{11}$$

where $p^k$ denotes the representation of diffusion step $k$.

**Encoder and Decoder:** Patches are processed by a series of Transformer encoders. Each encoder layer consists of multi-head self-attention and feed-forward networks. Finally, a flatten layer followed by a linear layer serves as the decoder, outputting the reconstructed $r_{1:L}^0$. Each encoder layer can be represented as follows:

$$r_{\text{PE}}^k = \text{AdaLN}(r_{\text{PE}}^k + \text{Multi\_Attn}(r_{\text{PE}}^k)) \in \mathbb{R}^{C \times N \times D}, \tag{12}$$

$$r_{\text{PE}}^k = \text{AdaLN}(r_{\text{PE}}^k + \text{Feed\_Forward}(r_{\text{PE}}^k)) \in \mathbb{R}^{C \times N \times D}. \tag{13}$$

# 4 EXPERIMENTS

## 4.1 DATA AND EXPERIMENTAL SETTING

**Datasets:** Six real-world MTS datasets are selected, each exhibiting distinct temporal dynamics, are selected: ETTm1, ETTm2, Weather, Solar-Energy, Electricity, and Traffic. Further details can be found in Appendix D.1.

**Metrics:** The performance of probabilistic forecasting is evaluated using the Continuous Ranked Probability Score (CRPS) and CRPS$_{\text{sum}}$, for both the proposed model and baseline models. Additionally, point forecasting performance is assessed using Mean Squared Error (MSE) and Mean Absolute Error (MAE). Detailed descriptions of these metrics can be found in Appendix E.

**Baselines:** Seven well-acknowledged long-term MTS forecasting models are carefully selected as baselines, including: (1) point forecasting methods: NSformer (Liu et al., 2022), TimesNet (Wu et al., 2023), DLinear (Zeng et al., 2023), PatchTST (Nie et al., 2023), and SparseVQ (Zhao et al., 2024); (2) probabilistic forecasting methods: TimeGrad (Rasul et al., 2021), CSDI (Tashiro et al., 2021), TimeDiff (Shen & Kwok, 2023), and TMDM (Li et al., 2024a). Detailed descriptions can be found in Appendix D.2.

**Implementation details:** In the experiments, the lookback window size $H$ and prediction length $L$ are set to 96 and 192, respectively. The diffusion process is configured with 100 steps, using a linear noise schedule where $\beta_1 = 10^{-4}$ and $\beta_K = 0.02$. A total of 100 samples are used to approximate the estimated distribution. All experiments are implemented using PyTorch (Paszke et al., 2019) and executed on an NVIDIA RTX A6000 48GB GPU. Further implementation details are provided in Appendix D.3. Unless otherwise stated, the point forecasting model used in the D$^3$U framework is SparseVQ.

## 4.2 RESULTS

### 4.2.1 MAIN RESULT

In this section, our method integrates SparseVQ (Zhao et al., 2024) and PatchDN within the D$^3$U framework, where SparseVQ functions as the conditioning network and PatchDN serves as the denoising network. Table 1 highlights the outstanding performance of our method in point forecasting, achieving a 28% improvement in MSE and a 21% improvement in MAE compared to the current state-of-the-art (SOTA) probabilistic MTS long-term forecasting method, TMDM (Li et al., 2024a). Additionally, our method's point forecasting capabilities are on par with the SOTA point forecasting models across four datasets. Compared to the SparseVQ model, our method demonstrates either superior or comparable point forecasting performance across all datasets, with particularly notable improvements in the Solar and Traffic datasets.

Table 1: Performance comparison on six real-world datasets based on MSE and MAE. The **best**/underline results are highlighted in **bold**/underline, respectively. Lower MSE and MAE values indicate better performance. *SparveVQ is used as the point forecasting model in the* $D^3U$ *(ours).*

| Model | Dataset | ETTm1 | | ETTm2 | | Weather | | Solar-Energy | | Electricity | | Traffic | |
|---|---|---|---|---|---|---|---|---|---|---|---|---|---|
| | Method | MSE | MAE | MSE | MAE | MSE | MAE | MSE | MAE | MSE | MAE | MSE | MAE |
| Point forecasting | NSformer(2022b) | 0.440 | 0.430 | 0.277 | 0.343 | 0.226 | 0.270 | 0.266 | 0.270 | 0.191 | 0.295 | 0.653 | 0.360 |
| | TimesNet(2023) | 0.374 | 0.387 | 0.249 | 0.309 | **0.219** | 0.261 | 0.296 | 0.318 | 0.184 | 0.289 | 0.617 | 0.336 |
| | DLinear(2023) | 0.380 | 0.389 | 0.284 | 0.362 | 0.237 | 0.296 | 0.320 | 0.398 | 0.196 | 0.285 | 0.598 | 0.370 |
| | PatchTST(2023) | 0.370 | 0.390 | 0.251 | 0.312 | 0.223 | 0.258 | 0.259 | 0.321 | 0.205 | 0.307 | 0.463 | 0.311 |
| | SparseVQ(2024) | **0.363** | **0.380** | 0.242 | **0.302** | 0.225 | 0.258 | 0.256 | 0.286 | 0.182 | 0.267 | 0.480 | 0.300 |
| | iTransformer(2024) | 0.377 | 0.391 | 0.250 | 0.309 | 0.221 | 0.254 | **0.233** | **0.261** | **0.164** | **0.255** | **0.418** | **0.284** |
| Probabilistic forecasting | TimeGrad(2021) | 1.716 | 1.057 | 1.385 | 0.732 | 0.885 | 0.551 | 1.211 | 1.004 | 0.645 | 0.723 | 0.932 | 0.807 |
| | CSDI(2021) | 0.867 | 0.690 | 1.291 | 0.576 | 0.842 | 0.523 | 0.848 | 0.818 | 0.553 | 0.795 | 0.921 | 0.678 |
| | TimeDiff(2023) | 0.796 | 0.577 | 0.284 | 0.342 | 0.277 | 0.331 | 1.169 | 0.936 | 0.730 | 0.690 | 1.465 | 0.851 |
| | TMDM(2024) | 0.607 | 0.558 | 0.524 | 0.493 | 0.244 | 0.286 | 0.295 | 0.317 | 0.222 | 0.329 | 0.721 | 0.411 |
| | ours | **0.363** | 0.386 | **0.241** | **0.302** | 0.222 | 0.264 | 0.237 | 0.270 | 0.179 | 0.267 | 0.468 | 0.299 |

Table 2 presents the superior probabilistic forecasting performance of our model, showing a 40% improvement in CRPS and a 5% improvement in CRPS$_{\text{sum}}$ compared to TMDM. In the subsequent ablation studies, we analyze the contributions of the $D^3U$ framework and the design of PatchDN to the improvements in long-term forecasting performance.

Table 2: Performance comparisons on six real-world datasets regarding CRPS and CRPS$_{\text{sum}}$. The **best**/underline results are highlighted in **bold**/underline. Lower CRPS and CRPS$_{\text{sum}}$ values indicate better performance. *SparveVQ is used as the point forecasting model in the* $D^3U$ *(ours).*

| Model | Dataset | ETTm1 | | ETTm2 | | Weather | | Solar-Energy | | Electricity | | Traffic | |
|---|---|---|---|---|---|---|---|---|---|---|---|---|---|
| | Method | CRPS | CRPS$_{\text{sum}}$ | CRPS | CRPS$_{\text{sum}}$ | CRPS | CRPS$_{\text{sum}}$ | CRPS | CRPS$_{\text{sum}}$ | CRPS | CRPS$_{\text{sum}}$ | CRPS | CRPS$_{\text{sum}}$ |
| Probabilistic Forecasting | TimeGrad(2021) | 0.665 | 0.996 | 0.785 | 1.051 | 0.482 | 0.503 | 0.783 | 1.167 | 0.503 | 1.452 | 0.657 | 1.683 |
| | CSDI(2021) | 0.773 | 0.852 | 0.625 | 0.782 | 0.508 | 0.465 | 0.649 | 0.681 | 0.465 | 0.823 | 0.612 | 1.275 |
| | TimeDiff(2023) | 0.454 | 0.846 | 0.316 | 0.180 | 0.293 | 0.400 | 0.900 | 1.164 | 0.475 | 0.594 | 0.671 | 0.823 |
| | TMDM(2024) | 0.429 | 0.633 | 0.380 | 0.226 | 0.226 | 0.292 | 0.375 | 0.267 | 0.446 | **0.137** | 0.552 | **0.179** |
| | ours | **0.285** | **0.574** | **0.243** | **0.141** | **0.207** | **0.283** | **0.186** | **0.266** | **0.202** | 0.160 | **0.232** | 0.186 |

### 4.2.2 ABLATION STUDY

To further demonstrate the advantages of the $D^3U$ framework, we modify the TMDM model to incorporate the $D^3U$ framework. Specifically, the NSformer (Liu et al., 2022) used in TMDM, which is originally co-trained with the denoising network, is replaced by a pre-trained NSformer model with identical parameter settings. In the $D^3U$ framework, the pre-trained NSformer predicts the deterministic component and provides guiding information, while the diffusion model's denoising network, responsible for modeling the residual distribution, employs an MLP network with consistent parameter settings. The experimental setup ensures full consistency between the training and evaluation processes. Table 3 highlights the superior performance of the $D^3U$ framework, where both point and probabilistic forecasting capabilities are significantly improved across all datasets compared to the original TMDM. TMDM models the overall data distribution, while the $D^3U$ framework explicitly decouples the deterministic and uncertain components of the data, requiring the modeling of only the high-uncertainty component. This reduces the complexity of the distribution and allows the DDPM to achieve stable probabilistic forecasting with fewer sampling steps (Appendix A).

**PatchDN design:** In all ablation experiments examining the design of PatchDN, SparseVQ serves as the conditioning network, providing guidance to the denoising network. In the comparative experiments, Table 4 shows that replacing the PatchDN denoising network in the $D^3U$ framework with the MLP network from TMDM or the UNet network from TimeDiff results in a decrease in both point and probabilistic forecasting performance. This demonstrates that PatchDN exhibits a stronger capability in modeling the high-uncertainty component.

Table 3: Performance promotion by applying $D^3U$ to TMDM.

| Mode | TMDM | | | | TMDM ($D^3U$) | | | |
|------|------|-----|------|----------------|------|-----|------|----------------|
| **Datasets** | MSE | MAE | CRPS | CRPS$_{sum}$ | MSE | MAE | CRPS | CRPS$_{sum}$ |
| ETTm1 | 0.607 | 0.558 | 0.429 | 0.633 | **0.441** | **0.432** | **0.324** | **0.616** |
| ETTm2 | 0.524 | 0.493 | 0.380 | 0.226 | **0.317** | **0.399** | **0.302** | **0.147** |
| Weather | 0.244 | 0.286 | 0.226 | 0.292 | **0.215** | **0.267** | **0.196** | **0.273** |
| Solar-Energy | 0.295 | 0.317 | 0.375 | 0.267 | **0.269** | **0.299** | 0.328 | **0.260** |
| Electricity | 0.222 | 0.329 | 0.446 | **0.137** | **0.216** | **0.328** | **0.381** | 0.157 |
| Traffic | 0.721 | 0.411 | 0.552 | **0.179** | **0.678** | **0.402** | **0.472** | 0.207 |

In PatchDN, two typical variants are designed for incorporating conditional information: the cross-attention method and the in-context method. Detailed descriptions of these two variants can be found in Appendix A. Table 4 shows that, compared to these two variants, the our method performs better in both point and probabilistic forecasting.

To further demonstrate the rationality behind the design of $D^3U$, two variants of the framework are created. The first variant uses the ground truth as the starting point for the conditional DDPM, i.e., $r^0 = y$. Experimental results show a significant decline in prediction performance. Compared to $D^3U$, modeling the entire data distribution using the same diffusion steps and sample sizes proves to be more challenging, resulting in degraded probabilistic forecasting performance and an inability to ensure accurate point forecasting. The second variant uses the final prediction output $\hat{y}$ of the conditioning network as guidance for the conditional DDPM. Experimental results also reveal a significant drop in prediction performance, suggesting that the encoder's output from a well-designed point forecasting model contains more beneficial information for guiding the denoising network.

Table 4: MSE, MAE and CRPS scores for different variants of the proposed method.

| Ablation Study | Mode $(f_\theta + g_\phi)$ | ETTm1 | | | Solar-Energy | | | Traffic | | |
|------|------|------|------|------|------|------|------|------|------|------|
| | | **MSE** | **MAE** | **CRPS** | **MSE** | **MAE** | **CRPS** | **MSE** | **MAE** | **CRPS** |
| **Denoise Network** | $SVQ + MLP^a$ | 0.372 | 0.396 | 0.294 | 0.330 | 0.313 | 0.242 | 0.525 | 0.331 | 0.297 |
| | $SVQ + UNet^b$ | 0.385 | 0.410 | 0.301 | 0.267 | **0.266** | 0.219 | 0.469 | 0.301 | 0.289 |
| **Structure Design** | $SVQ + PatchDN(CAttn)^c$ | 0.370 | 0.390 | 0.295 | 0.237 | 0.281 | 0.193 | 0.483 | 0.307 | 0.240 |
| | $SVQ + PatchDN(InC)^d$ | 0.366 | **0.385** | 0.290 | 0.238 | 0.271 | 0.187 | 0.486 | 0.315 | 0.244 |
| **Framework Design** | $SVQ + PatchDN(All)^e$ | 0.859 | 0.699 | 0.516 | 0.348 | 0.302 | 0.251 | 0.687 | 0.396 | 0.302 |
| | $SVQ + PatchDN(\hat{y})^f$ | 0.408 | 0.421 | 0.312 | 0.259 | 0.301 | 0.241 | 0.479 | 0.308 | 0.251 |
| | **Ours** | **0.361** | **0.385** | **0.284** | **0.236** | 0.270 | **0.186** | **0.468** | **0.299** | **0.232** |

[1] SVQ is the abbreviation for SparseVQ.
[2] $a$ means the MLP serves as the denoising network in the TMDM model and consists of four linear layers; $b$ means the UNet, used as the denoising network in the TimeDiff model, is built using a convolutional neural network-based UNet architecture.
[3] $c$ marks PatchDN based on the cross-attention method; $d$ marks PatchDN based on the in-context method.
[4] $e$ marks the framework variant that models the entire data distribution; $f$ marks the framework variant that employs $\hat{y}$ as the guidance.

**Framework generality:** The $D^3U$ framework is designed as a plug-and-play solution that can be seamlessly integrated into existing point forecasting models. The conditioning network is replaced with NSformer, PatchTST, and SparseVQ, respectively. As shown in Table 5, applying the $D^3U$ framework improves each model's point forecasting capabilities, while also providing them with probabilistic forecasting abilities.

Table 5: Performance promotion by applying the proposed framework to point forecasting models.

| Dataset | ETTm1 | | | Solar-Energy | | | Traffic | | |
|------|------|------|------|------|------|------|------|------|------|
| **Method** | MSE | MAE | CRPS | MSE | MAE | CRPS | MSE | MAE | CRPS |
| NSformer | 0.440 | 0.430 | – | **0.266** | **0.270** | – | **0.653** | **0.360** | – |
| NSformer ($D^3U$) | **0.436** | **0.427** | 0.317 | 0.268 | 0.272 | 0.202 | 0.657 | 0.367 | 0.284 |
| PatchTST | **0.370** | **0.390** | – | 0.259 | 0.321 | – | 0.463 | 0.311 | – |
| PatchTST ($D^3U$) | 0.387 | 0.405 | 0.299 | **0.233** | **0.281** | 0.221 | **0.452** | **0.297** | 0.234 |
| SparseVQ | 0.363 | **0.380** | – | 0.256 | 0.286 | – | 0.480 | **0.300** | – |
| SparseVQ ($D^3U$) | **0.361** | 0.385 | 0.284 | **0.237** | **0.270** | 0.185 | **0.475** | 0.309 | 0.232 |

[1] – means that point forecasting models do not have probabilistic forecasting abilities. The CRPS value degrades to the Normalized Mean Square Error (NMAE), which is omitted here.

### 4.3 RETHINK OF UNCERTAINTY MODELING IN $D^3U$

In the previous two sections, extensive experiments have validated the effectiveness of the $D^3U$ framework. However, some results in Table 2 and Table 3 suggest that $D^3U$ may face challenges in uncertainty modeling when applied to MTS data with extremely high variable dimensions (e.g., Electricity, Traffic), particularly reflected in $CRPS_{sum}$. This issue is related to the composition of uncertainty in the prediction error. As discussed in Section 2.1, the prediction error of the point forecasting model primarily consists of epistemic uncertainty and aleatoric uncertainty (Gawlikowski et al., 2023). In practice, a relatively simple yet effective approach is adopted by approximating the high-uncertainty components using the prediction error, $y - \hat{y}$. However, it is observed that higher epistemic uncertainty tends to result in wider probabilistic prediction intervals, which subsequently impacts the model's performance as measured by $CRPS_{sum}$. As shown in Fig. 3, when the point

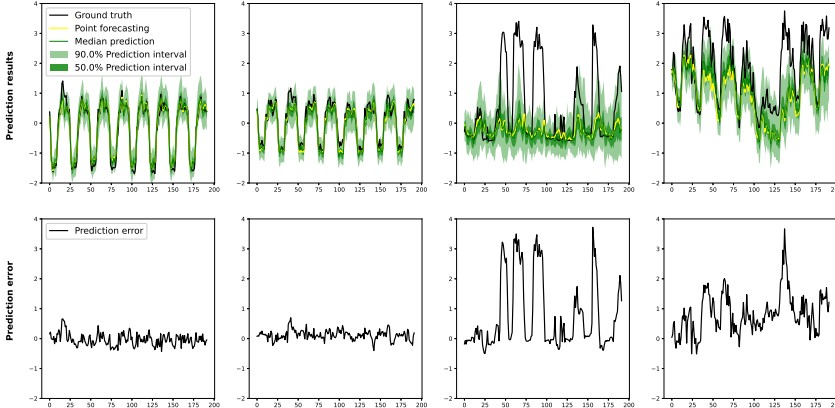

Figure 3: Cases of probabilistic prediction intervals with different uncertainty compositions.

forecasting model effectively models the deterministic components, the prediction error is mainly composed of aleatoric uncertainty, resulting in more accurate prediction intervals for probabilistic forecasting. Conversely, epistemic uncertainty becomes dominant when the point forecasting model struggles with the deterministic components, leading to wider prediction intervals. Therefore, employing a more powerful point forecasting model can effectively mitigate this issue. As shown in Table 6, after replacing the conditioning network from SparseVQ with iTransformer, the $D^3U$ framework achieved significant improvement in the $CRPS_{sum}$ metric. This also suggests that a potential direction for improving the $D^3U$ framework in future work would be to explore better methods for separating and addressing epistemic uncertainty and aleatoric uncertainty.

Table 6: Performance comparison of different point forecasting models employed by the $D^3U$ framework.

| Mode | Electricity | | | | Traffic | | | |
|---|---|---|---|---|---|---|---|---|
| | MSE | MAE | CRPS | $CRPS_{sum}$ | MSE | MAE | CRPS | $CRPS_{sum}$ |
| SparseVQ + PatchDN | 0.179 | 0.267 | 0.202 | 0.160 | 0.469 | 0.299 | 0.232 | 0.186 |
| iTransformer + PatchDN | **0.168** | **0.261** | **0.195** | **0.151** | **0.421** | **0.290** | **0.222** | **0.169** |

## 5 CONCLUSION

In this paper, we introduce the $D^3U$ framework for probabilistic long-term MTS forecasting. $D^3U$ decouples the deterministic and uncertain components of time series, leveraging pre-trained point forecasting models to model the high-certainty components, while employing a conditional DDPM to perform probabilistic forecasting on the high-uncertainty components. As a plug-and-play framework, $D^3U$ can be seamlessly integrated into existing point forecasting models to enable probabilistic forecasting. To better capture the high-uncertainty components, PatchDN is designed within the conditional DDPM. Comprehensive experiments on six real-world datasets demonstrate the outstanding probabilistic forecasting performance and competitive point forecasting capability of $D^3U$.

ACKNOWLEDGMENTS

We appreciate constructive feedback from anonymous reviewers and meta-reviewers. This work is supported by the National Natural Science Foundation of China under Grant No.61971057 and National Key R&D Program of China under Grant No. 2020YFC1522503.

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

## A  APPENDIX: EXPERIMENTS

**Details of PatchDN Variants:** 1. Cross-attention method: conditional information serve as keys and values in the attention mechanism, guiding the model to adaptively model the input patches (Li et al., 2024b). 2. In-context method: the conditional information is treated as part of the input sequence to the Transformer encoder, processed alongside other patch embeddings (Bao et al., 2023).

In Table 4, we compare the performance of our method with the two structure variants. However, the second-best results in this experiment are also achieved by these two variants, prompting further analysis of the FLOPs and inference time for all three mechanisms. Table 7 demonstrates that our method has lower computational complexity and inference time compared to the other two, which explains its superior overall performance.

Table 7: Comparison of FLOPs and Inference Time with the Two Variants (100 samples).

| Mode | ETTm1 | | Solar-Energy | | Traffic | |
|---|---|---|---|---|---|---|
| | #FLOPS (G) | Inference time (min) | #FLOPS (G) | Inference time (min) | #FLOPS (G) | Inference time (min) |
| SVQ + PatchDN(CAttn)[a] | 4.933 | 0.094 | 20.69 | 0.446 | 130.0 | 0.452 |
| SVQ + PatchDN(InC)[b] | 4.632 | 0.065 | 20.94 | – | 131.7 | 0.336 |
| **Ours** | **3.569** | **0.050** | **15.74** | **0.238** | **99.03** | **0.299** |

[1] SVQ is the abbreviation for SparseVQ.
[2] a marks PatchDN based on the cross-attention method; b marks PatchDN based on the in-context method.
[3] − indicates an out-of-memory (OOM) error on a 48GB GPU.

**Diffusion Steps:** Fig. 4 illustrates the sensitivity of three time series probabilistic forecasting models to diffusion steps. Due to the decoupling advantage of the $D^3U$ framework, which reduces the

complexity of distribution modeling, the figure shows that PatchDN achieves optimal performance at 70 steps on the ETTm1 dataset and 50 steps on the Weather dataset. Compared to other models, PatchDN exhibits lower sensitivity to diffusion steps.

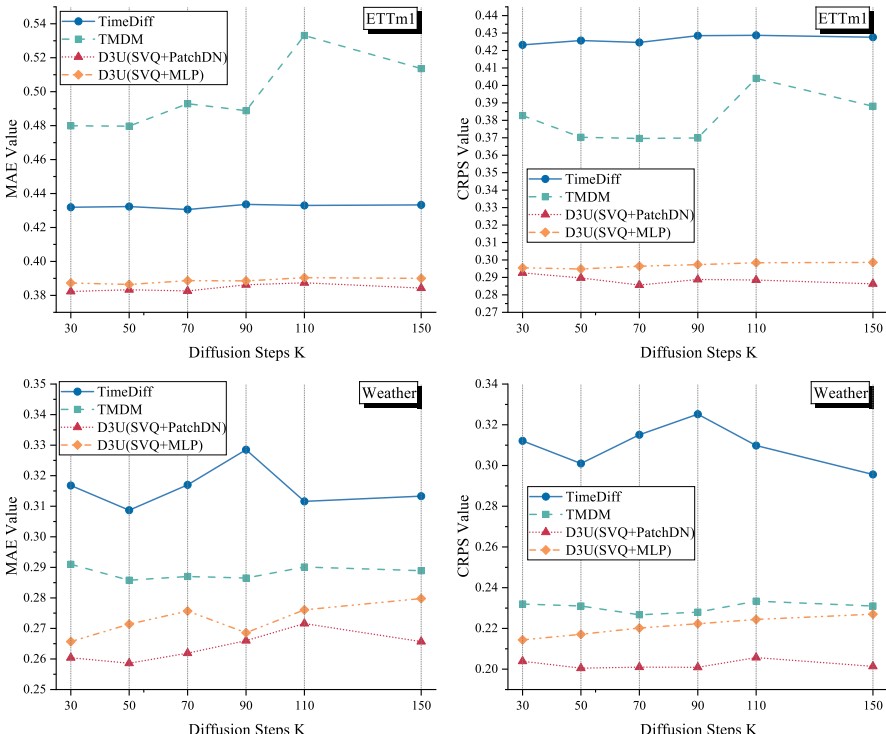

Figure 4: The impact of different number of diffusion steps $K$.

**Prediction Strategy:** Table 8 presents a comparison of different prediction strategies. In low-dimensional datasets, the strategy of predicting $\epsilon_\phi$ typically performs better, whereas in high-dimensional datasets, the strategy of predicting $r_\phi$ tends to be more effective. This may be due to the fact that high-dimensional time series datasets contain more highly irregular noise components, making it more challenging to estimate the diffusion noise $\epsilon_\phi$.

Table 8: Quantitative comparison of different prediction strategy.

| Mode | ETTm1 | | | ETTm2 | | | Weather | | | Solar-Energy | | | Electricity | | | Traffic | | |
|---|---|---|---|---|---|---|---|---|---|---|---|---|---|---|---|---|---|---|
| | MSE | MAE | CRPS | MSE | MAE | CRPS | MSE | MAE | CRPS | MSE | MAE | CRPS | MSE | MAE | CRPS | MSE | MAE | CRPS |
| $r_\phi$ | **0.363** | 0.386 | **0.285** | 0.251 | 0.318 | 0.243 | 0.227 | 0.279 | 0.207 | **0.237** | **0.270** | **0.186** | **0.179** | 0.267 | 0.202 | **0.469** | 0.299 | **0.229** |
| $\epsilon_\phi$ | 0.363 | **0.382** | 0.286 | **0.241** | **0.302** | **0.236** | **0.222** | **0.264** | **0.203** | 0.254 | 0.276 | 0.186 | 0.184 | 0.269 | **0.201** | 0.481 | **0.298** | 0.229 |

# B APPENDIX: SERIES DECOMPOSITION

## B.1 SERIES DECOMPOSITION

In this paper, a moving average is applied to the input sequence to obtain the component $X_T$, which contains trend information. The amount of trend information retained in $X_T$ is controlled by the moving average kernel $k$ (Wu et al., 2021; Zeng et al., 2023):

$$X_T = \text{AvgPool}(X; k) \tag{14}$$

For the seasonal component, a Fourier layer based on Fourier bases is employed (Yuan & Qiao, 2024). The number of retained frequency components is controlled by the Fourier factor $f$:

$$X_S = \text{Layer\_FFT}(X - X_T; f) \tag{15}$$

The residual component is defined as:

$$X_R := X - X_T - X_S \tag{16}$$

## B.2 Series Decomposition Experiments

As mentioned in Section 3, the input of the conditional network, $x_h$, can either use the original time series $x$ or the $x_T + x_S$. Table 9 presents the impact of different kinds of $x_h$ on the forecasting performance of two point forecasting models (NSformer, PatchTST) and their $D^3U$ versions.

Table 9: The impact of different types of $x_h$ on the point forecasting model and its $D^3U$ version's forecasting performance. 'All' means using the original series $x$ as input. 'K* F*' means using the $x_T + x_S$ as input. For example, 'K3' denotes a moving average kernel $k = 3$ in Eq. 14 and 'F2' denotes a Fourier Factor $f = 2.0$ in Eq. 15. In this experiment, the lookback window size $H$ is 96 and the prediction length $L$ is 192.

| Dataset | Type of $x_h$ Metric | All MSE | MAE | CRPS | K3 F2 MSE | MAE | CRPS | K7 F2 MSE | MAE | CRPS | K7 F1 MSE | MAE | CRPS | K15 F1 MSE | MAE | CRPS |
|---|---|---|---|---|---|---|---|---|---|---|---|---|---|---|---|---|
| | NSformer | 0.440 | 0.430 | — | **0.435** | **0.430** | — | 0.438 | 0.431 | — | 0.439 | 0.430 | — | 0.448 | 0.432 | — |
| | NSformer($D^3$U) | 0.436 | 0.427 | 0.317 | 0.436 | 0.431 | 0.321 | 0.429 | **0.426** | 0.317 | **0.428** | 0.427 | **0.315** | 0.453 | 0.432 | 0.323 |
| ETTm1 | PatchTST | **0.370** | **0.390** | — | 0.374 | 0.393 | — | 0.375 | 0.393 | — | 0.375 | 0.393 | — | 0.374 | 0.392 | — |
| | PatchTST($D^3$U) | 0.387 | 0.405 | 0.299 | 0.381 | 0.400 | 0.294 | 0.386 | 0.403 | 0.295 | 0.385 | 0.402 | 0.296 | **0.376** | **0.394** | **0.289** |
| | NSformer | 0.226 | 0.270 | — | **0.218** | **0.263** | — | 0.225 | 0.268 | — | 0.227 | 0.270 | — | 0.232 | 0.272 | — |
| | NSformer($D^3$U) | 0.233 | 0.281 | 0.209 | **0.226** | **0.275** | **0.207** | 0.237 | 0.283 | 0.212 | 0.245 | 0.282 | 0.213 | 0.240 | 0.284 | 0.212 |
| Weather | PatchTST | **0.223** | **0.258** | — | 0.224 | **0.258** | — | 0.224 | 0.259 | — | 0.225 | 0.259 | — | 0.226 | 0.260 | — |
| | PatchTST($D^3$U) | 0.236 | 0.293 | 0.218 | **0.224** | 0.267 | **0.205** | 0.236 | 0.288 | 0.215 | **0.224** | 0.267 | 0.206 | 0.225 | **0.264** | 0.206 |

## C    Appendix: Training Strategy Experiments

The D3U framework freezes the parameters of the pre-trained point forecasting model, which reduces the number of trainable parameters and the overall training time. To demonstrate this advantage, we conducted experiments with three additional training configurations: (1)$D^3U$ with series decomposition (DCP), denoted as $D^3U$-DCP: the inputs of the point prediction model and the diffusion model are $x_T + x_S$ and $x_R := x - x_T + x_S$, respectively. (2) $D^3U$-FT: fine-tuning the pre-trained point forecasting model while training the diffusion model, and (3) $D^3U$-FS: training the entire model from scratch without using the pre-trained point forecasting model. The details of the experimental results are shown in Table 10. The results indicate that, on most datasets, the fine-tuned and from-scratch training configurations do not show significant differences compared to the original $D^3U$. However, the training approach using the frozen pre-trained model results in the minimum number of training parameters and the shortest training time. In these three schemes, $D^3U$-DCP performs the worst. We think the reason for this result is that DCP provides a coarse approach to separating the high-certainty and high-uncertainty components of time series data. As shown in the experimental results of Table 9, the effectiveness of DCP is highly dependent on the choice of hyperparameters $k$ and $f$. Therefore, we do not consider DCP a reliable approach for extracting high-certainty or high-uncertainty components in all cases. In the setup we adopt in the paper, the undecomposed data $x$ is used as the input to the point forecasting model, and the prediction error $y - \hat{y}$ from the point forecasting model is used as the input to the diffusion model. As analyzed in Section 2.1, the prediction error of the point forecasting model primarily consists of two types of uncertainty: aleatoric uncertainty and epistemic uncertainty. Thus, the prediction error is a reasonable approximation of the high-uncertainty component in the time series data that is difficult for the point forecasting model to capture. Details of the number of training parameters and the training time for the different training modes are presented in Table 11.

## D    Appendix: implementation details

### D.1    bencnmark datasets

For our experiments, we use `ETTm1`, `ETTm2`, `Weather`, `Solar-Energy`, `Electricity` and `Traffic` open-source datasets, with their properties listed in Table 12. Following the methodolo-

Table 10: MSE, MAE and CRPS scores for different training framework variants of the proposed method. $D^3U$-DCP: The inputs to the point prediction model and the diffusion model are $x_T + x_S$ and $x_R := x - x_T + x_S$, respectively. $D^3U$-FT: Fine-tuning the point forecasting model while training the diffusion model. $D^3U$-FS: Training the entire model from scratch.

| Training Mode | ETTm1 | | | ETTm2 | | | Weather | | | Solar-Energy | | | Electricity | | | Traffic | | |
|---|---|---|---|---|---|---|---|---|---|---|---|---|---|---|---|---|---|---|
| | MSE | MAE | CRPS | MSE | MAE | CRPS | MSE | MAE | CRPS | MSE | MAE | CRPS | MSE | MAE | CRPS | MSE | MAE | CRPS |
| $D^3U$-DCP | 0.395 | 0.405 | 0.330 | 0.251 | 0.312 | 0.265 | 0.228 | 0.268 | 0.240 | 0.294 | 0.324 | 0.272 | 0.275 | 0.377 | 0.280 | 0.790 | 0.510 | 0.391 |
| $D^3U$-FT | 0.368 | 0.388 | 0.286 | 0.253 | 0.322 | 0.247 | 0.222 | 0.271 | 0.202 | 0.217 | 0.259 | 0.169 | 0.174 | 0.266 | 0.197 | 0.535 | 0.292 | 0.224 |
| $D^3U$-FS | 0.364 | 0.386 | 0.284 | 0.248 | 0.317 | 0.243 | 0.230 | 0.284 | 0.210 | 0.220 | 0.262 | 0.172 | 0.177 | 0.266 | 0.199 | 0.516 | 0.304 | 0.233 |
| $D^3U$ | 0.363 | 0.386 | 0.285 | 0.241 | 0.302 | 0.243 | 0.222 | 0.264 | 0.207 | 0.237 | 0.270 | 0.186 | 0.179 | 0.267 | 0.202 | 0.468 | 0.299 | 0.232 |

Table 11: Trainable parameters and training time for different training framework variants of the proposed method. $D^3U$-DCP: The inputs to the point prediction model and the diffusion model are $x_T + x_S$ and $x_R := x - x_T + x_S$, respectively. $D^3U$-FT: Fine-tuning the point forecasting model while training the diffusion model. $D^3U$-FS: Training the entire model from scratch.

| Training Mode | ETTm1 | | ETTm2 | | Weather | | Solar-Energy | | Electricity | | Traffic | |
|---|---|---|---|---|---|---|---|---|---|---|---|---|
| | Trainable Parameters (M) | Training time (s/batch) | Trainable Parameters (M) | Training time (s/batch) | Trainable Parameters (M) | Training time (s/batch) | Trainable Parameters (M) | Training time (s/batch) | Trainable Parameters (M) | Training time (s/batch) | Trainable Parameters (M) | Training time (s/batch) |
| $D^3U$-DCP | 1.659 | 4.972 | 1.659 | 4.962 | 3.024 | 14.069 | 1.322 | 79.627 | 2.166 | 204.020 | 3.495 | 319.429 |
| $D^3U$-FT | 7.380 | 7.471 | 7.380 | 7.493 | 4.825 | 18.106 | 3.504 | 105.355 | 10.249 | 306.249 | 6.467 | 341.098 |
| $D^3U$-FS | 7.380 | 7.552 | 7.380 | 7.471 | 4.825 | 18.050 | 3.504 | 105.139 | 10.249 | 306.112 | 6.467 | 340.347 |
| $D^3U$ | 1.659 | 4.567 | 1.659 | 4.621 | 3.024 | 13.712 | 1.322 | 76.627 | 2.166 | 198.587 | 3.495 | 316.330 |

gies of Wu et al. (2021) and Zhou et al. (2022), the datasets are split chronologically into training, validation, and test sets. A 6:2:2 ratio is used for ETTm1 and ETTm2, while a 7:1:2 ratio is applied for Weather, Solar-Energy, Traffic, and Electricity. The dataset can be obtained through the links below.

 (i) ETTm1,ETTm2: https://github.com/zhouhaoyi/ETDataset.
 (ii) Weather: https://www.bgc-jena.mpg.de/wetter/.
 (iii) Solar-Energy:https://archive.ics.uci.edu/ml/datasets/ElectricityLoadDiagrams20112014.
 (iv) Electricity:https://archive.ics.uci.edu/ml/datasets/ElectricityLoadDiagrams20112014.
 (v) Traffic: http://pems.dot.ca.gov/

Table 12: Detailed information of the datasets used in our benchmark, including data frequency, number of time series (dimension), context length, and prediction length. Dataset size indicates the total number of time points in the train, validation, and test splits, respectively.

| Dataset | Dimension | Frequency | Dataset Size | Context length | Prediction length |
|---|---|---|---|---|---|
| ETTm1,ETTm2 | 7 | 15 Min | (34465, 11521, 11521) | 96 | 192 |
| Weather | 21 | 10 Min | (36792, 5271, 10540) | 96 | 192 |
| Solar-Energy | 137 | 10 Min | (36601, 5161, 10417) | 96 | 192 |
| Electricity | 321 | 1 Hour | (18317, 2633, 5261) | 96 | 192 |
| Traffic | 862 | 1 Hour | (12185, 1757, 3509) | 96 | 192 |

### D.2 BASELINES IN MAIN EXPERIMENTS

Seven well-acknowledged long-term MTS forecasting models are carefully selected as baselines, including: (1) point forecasting methods: NSformer (Liu et al., 2022), TimesNet (Wu et al., 2023), DLinear (Zeng et al., 2023), PatchTST (Nie et al., 2023), and SparseVQ (Zhao et al., 2024); (2) probabilistic forecasting methods: TimeGrad (Rasul et al., 2021), CSDI (Tashiro et al., 2021), TimeDiff (Shen & Kwok, 2023), and TMDM (Li et al., 2024a).

The code and descriptions of the baseline methods can be obtained from the following sources.

(i) NSformer: a novel framework that enhances Transformer models for time series forecasting by integrating Series Stationarization to unify input statistics and De-stationary Attention to recover intrinsic non-stationary information.

Code: `https://github.com/thuml/Nonstationary_Transformers`

(ii) TimesNet: a task-general backbone for time series analysis that transforms 1D time series into 2D tensors to effectively model complex temporal variations through adaptive multi-periodicity discovery and a parameter-efficient inception block.

Code: `https://github.com/thuml/TimesNet`

(iii) DLinear: a simple one-layer linear model for long-term time series forecasting that out-performs sophisticated Transformer-based models by effectively preserving temporal relations.

Code: `https://github.com/cure-lab/LTSF-Linear`

(iv) PatchTST: an efficient Transformer-based model for multivariate time series forecasting that utilizes segmentation into subseries-level patches and channel independence.

Code: `https://github.com/yuqinie98/patchtst`

(v) SparseVQ: a novel approach for time series analysis that utilizes sparse vector quantization and Reverse Instance Normalization to address distribution shifts and noise, replacing the Feed-Forward layer to enhance computational efficiency and reduce overfitting.

Code: `https://anonymous.4open.science/r/Sparse-VQ-DC28`

(vi) TimeGrad: an autoregressive model for multivariate probabilistic time series forecasting which samples from the data distribution at each time step by estimating its gradient.

Code: `https://github.com/microsoft/ProbTS`

(vii) CSDI: a novel time series imputation method that utilizes score-based diffusion models conditioned on observed data.

Code: `https://github.com/microsoft/ProbTS`

(viii) TimeDiff: a non-autoregressive diffusion model for time series prediction that leverages innovative conditioning mechanisms—future mixup and autoregressive initialization.

Code: There is no publicly available code; we obtained the code by emailing the author.

(ix) TMDM: combines conditional diffusion processes with transformers to enable precise distribution forecasting in multivariate time series while effectively incorporating uncertainty.

Code: `https://github.com/LiYuxin321/TMDM`

## D.3   IMPLEMENTATION DETAILS

To accelerate the inference of the DDPM, DPM-Solver (Lu et al., 2022) is employed, allowing the number of denoising steps to be empirically reduced to fewer than 20. The proposed model is trained using the Adam optimizer with a learning rate of $10^{-4}$. Following the parameter settings outlined in Li et al. (2024a), early stopping is applied after 15 epochs without improvement, with a maximum of 100 training epochs. Table 13 presents the hyperparameters of the SparseVQ+PatchDN model for both training and testing across each dataset. The conditioning network is the SparseVQ model, with the primary hyperparameters being `encoder_layers`, `d_model`, `d_ff`, `num_codebook`, and `codebook_size`. The hyperparameters for PatchDN are consistent across all datasets, consisting of a single-layer Transformer encoder with 8 heads and a latent space dimension of 128.

Table 13: Hyper-parameter values for the SparseVQ+PatchDN Model.

| Dataset | Diffusion Train batch_size | Diffusion Test batch_size | Condition network encoder_layers | Condition network d_model | Condition network d_ff | Condition network num_codebook | Condition network codebook_size |
|---|---|---|---|---|---|---|---|
| ETTm1,ETTm2 | 128 | 64 | 2 | 512 | 512 | 1 | 256 |
| Weather | 128 | 64 | 2 | 256 | 512 | 1 | 256 |
| Solar-Energy | 64 | 16 | 2 | 256 | 512 | 1 | 1000 |
| Electricity | 64 | 8 | 2 | 512 | 512 | 2 | 256 |
| Traffic | 16 | 2 | 3 | 256 | 256 | 2 | 512 |

# E APPENDIX: METRICS

## E.1 METRICS FOR POINT FORECASTING

**Mean Squared Error (MSE).** MSE is calculated as the average of the squared differences between the predicted values and the actual values, defined mathematically as:

$$\text{MSE} = \frac{1}{C \times L} \sum_{c=1}^{C} \sum_{l=1}^{L} (r_l^c - \hat{r}_l^c)^2, \tag{17}$$

where $C$ represents the number of variates, $L$ denotes the length of the series, and $r_l^c$ and $\hat{r}_l^c$ indicate the ground-truth value and the predicted value, respectively.

**Mean Absolute Error (MAE).** MAE measures the average magnitude of the errors in a set of predictions, without considering their direction. It is calculated as:

$$\text{MAE} = \frac{1}{C \times L} \sum_{c=1}^{C} \sum_{l=1}^{L} |r_l^c - \hat{r}_l^c|. \tag{18}$$

## E.2 METRICS FOR PROBABILISTIC FORECASTING

**Continuous Ranked Probability Score (CRPS).** The CRPS (Matheson & Winkler, 1976) quantifies the difference between the cumulative distribution function (CDF) $F$ of the predicted probabilities and the CDF of the observed outcomes $r$, represented as:

$$\text{CRPS} = \int_{\mathbb{R}} (F(z) - \mathbb{I}_{\{r \leq z\}})^2 \, dz, \tag{19}$$

where $\mathbb{I}_{\{r \leq z\}}$ is the indicator function that equals one if $r \leq z$ and zero otherwise. As a proper scoring function, CRPS achieves its minimum when the predictive distribution $F$ matches the true data distribution. Using the empirical CDF $\hat{F}(z) = \frac{1}{n} \sum_{i=1}^{n} \mathbb{I}\{R_i \leq z\}$, CRPS can be computed from simulated samples of the conditional distribution $p_\theta(r_t|h_t)$.

**CRPS$_{\text{sum}}$.** CRPS$_{\text{sum}}$ extends CRPS to multivariate time series. Specifically, CRPS$_{\text{sum}}$ is defined as:

$$\text{CRPS}_{\text{sum}} = \mathbb{E}_t[\text{CRPS}(F_{\text{sum}}^{-1}, \sum_i r_i^t)], \tag{20}$$

where $F_{\text{sum}}^{-1}$ is obtained by summing the samples across dimensions and then ordering them to derive the quantiles.

# F APPENDIX: FORECAST SHOWCASES

## F.1 CASE STUDY

To demonstrate the superiority of the proposed method, Fig. 5 visualizes the ground truth and predictions of time series across two dimensions in the ETTm1 and Weather datasets, along with the 50% and 90% prediction intervals. The results show that the proposed method achieves higher point forecasting accuracy (median prediction, represented by the dark green line) compared to the other two models, and demonstrates a stronger capability in estimating the distribution of the time series.

## F.2 DENOISING PROCESS SHOWCASES

To clearly demonstrate the generation process of time series data, Fig. 6, Fig. 7 and Fig. 8 illustrate the denoising process (aka reverse process) of time series data across two dimensions in the ETTm1, Electricity, and Traffic datasets, from diffusion step 100 (start point of denoising process) to step 0 (end point of denoising process). The results indicate that at diffusion step 100, point prediction models can provide prior knowledge to diffusion models, ensuring predictions oscillate near the true values rather than gradually denoising from pure Gaussian noise. This effectively reduces the difficulty of denoising. As the steps decrease, the width of the 50% and 90% prediction intervals gradually narrows, achieving accurate point prediction precision at step 0 (denoted by the median prediction in dark green lines) and effectively estimating the distribution of time series data.

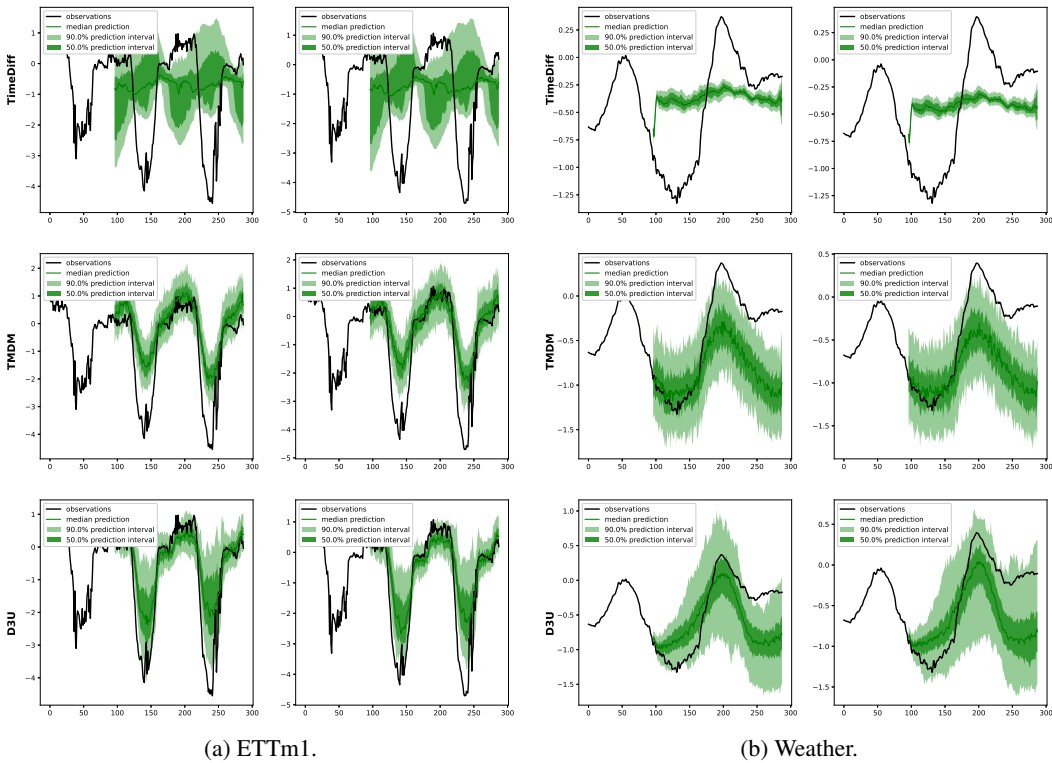

(a) ETTm1.  (b) Weather.

Figure 5: More comparison of prediction intervals for the ETTm1 and Weather Datasets. The black line representing the test set ground-truth.

### F.3  PREDICTION RESULTS VISUALIZATION

To further illustrate the predictive performance of the D3U framework, Fig. 9, Fig. 10, Fig. 11, Fig. 12, Fig. 13, Fig. 14 and Fig. 15 present the outputs of the point prediction model, the probabilistic prediction model (provided by the mean of its samples), the overall prediction results of the framework, and the corresponding ground truth values.

These figures show that incorporating the results of probabilistic prediction enhances the overall accuracy of the model's predictions. This improvement stems from the D3U framework's diffusion model effectively capturing the distribution of components with high-uncertainty in the data. Additionally, the mean of the diffusion model's samples, denoted as $\mathrm{Avg}(\hat{y}_g)$, exhibits noticeable periodicity. Visualizing these samples further reinforces this conclusion. If we treat the time series as a signal, the true distribution of the prediction error $p(y - \hat{y}|t)$ is inherently a time-dependent random variable, i.e., a stochastic process. Therefore, if the diffusion model successfully learns the distribution of $p(y - \hat{y}|t)$, it should also model an inherently time-dependent distribution. Fig. 9b, Fig. 10b, Fig. 11b, Fig. 12b, Fig. 13b, Fig. 14b and Fig. 15b illustrate 100 samples generated from the distribution learned by the diffusion model. The range covered by these samples reflects the uncertainty of the data—the wider the range, the higher the uncertainty. It is evident that the range of coverage exhibits clear periodicity over time and narrows significantly at time steps with lower prediction errors (lower uncertainty). This demonstrates that within the D3U framework, the diffusion model effectively captures the distribution of prediction errors rather than merely introducing time-independent randomness into the point prediction results.

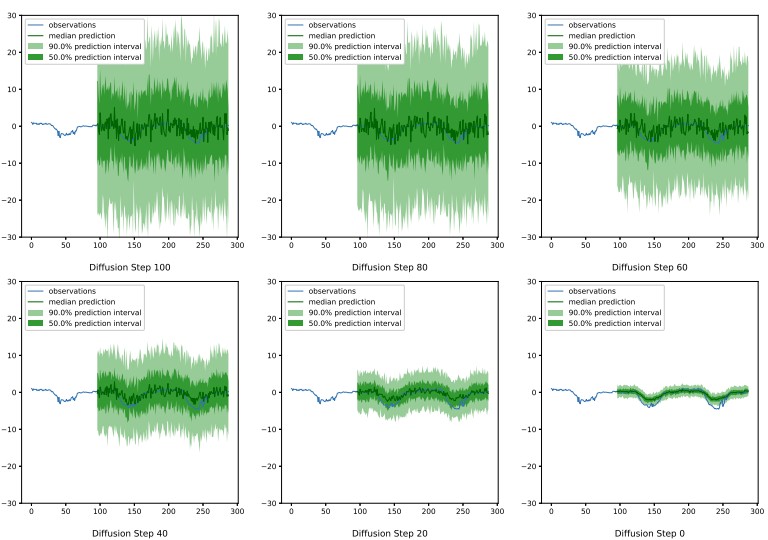

(a) ETTm1 $0^{th}$ dimension.

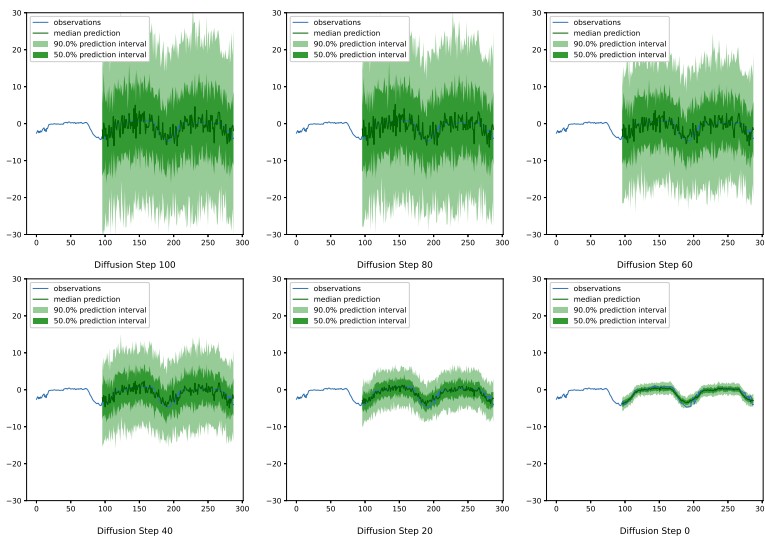

(b) ETTm1 $2^{th}$ dimension.

Figure 6: Visualization of the Denoising Process of ETTm1 from Diffusion Step 100 to Step 0.

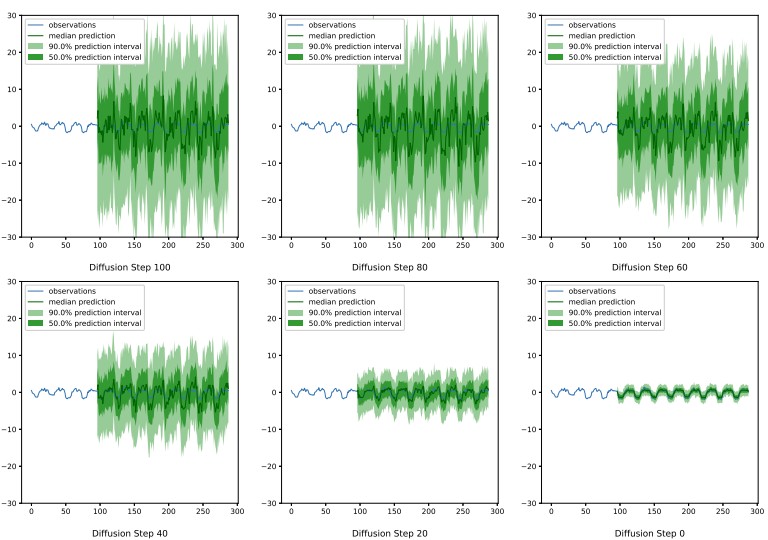

(a) Electricity $138^{th}$ dimension.

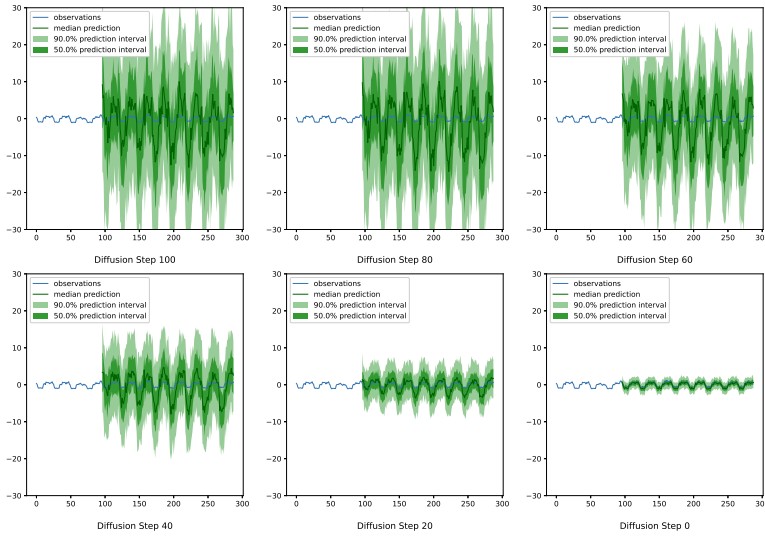

(b) Electricity $139^{th}$ dimension.

Figure 7: Visualization of the Denoising Process of Electricity from Diffusion Step 100 to Step 0.

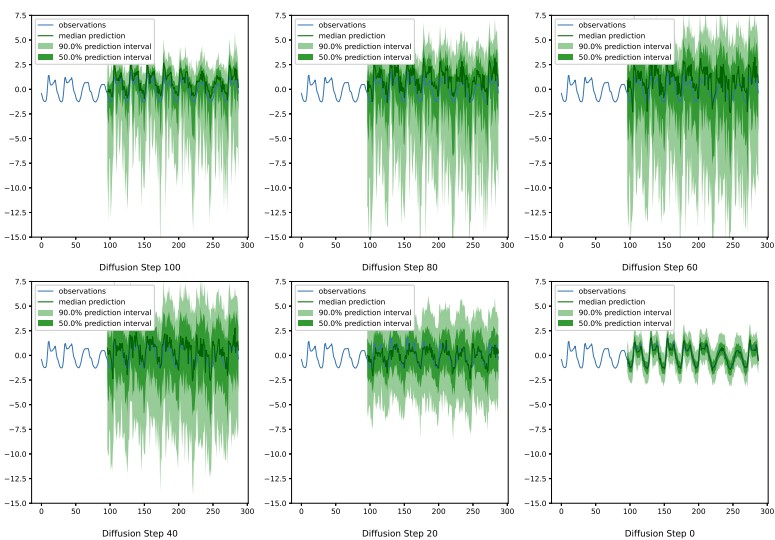

(a) Traffic $300^{th}$ dimension.

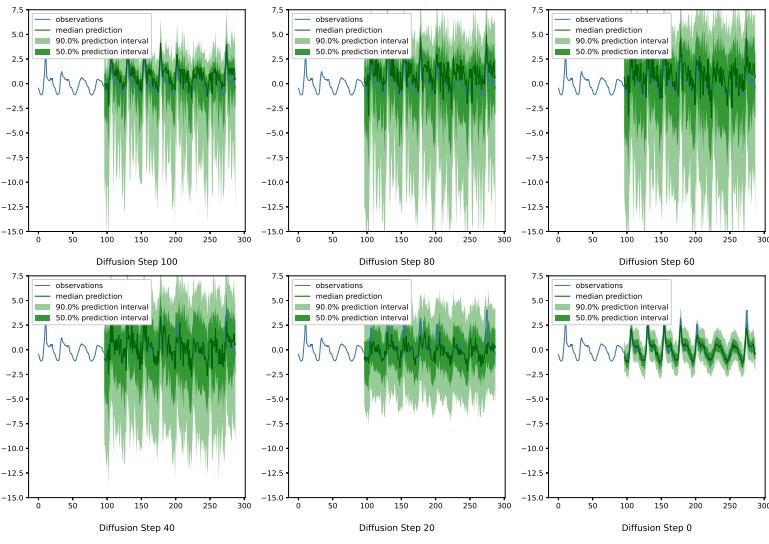

(b) Traffic $410^{th}$ dimension.

Figure 8: Visualization of the Denoising Process of Traffic from Diffusion Step 100 to Step 0.

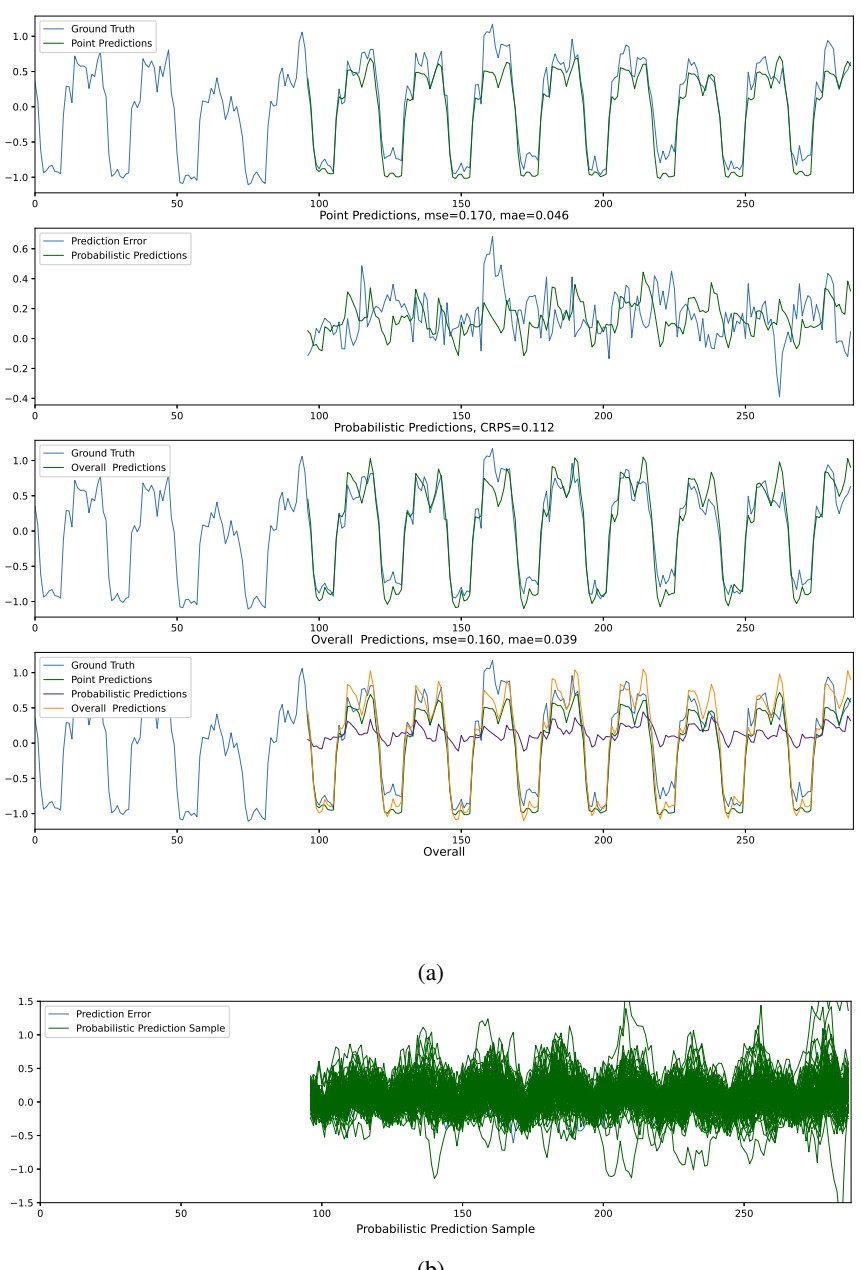

(a)

(b)

Figure 9: Visualization of the Predictions of Eletricity ($139^{th}$ dimension). This case illustrates a scenario with relatively low epistemic uncertainty. Probabilistic prediction: the mean of the diffusion model's samples.

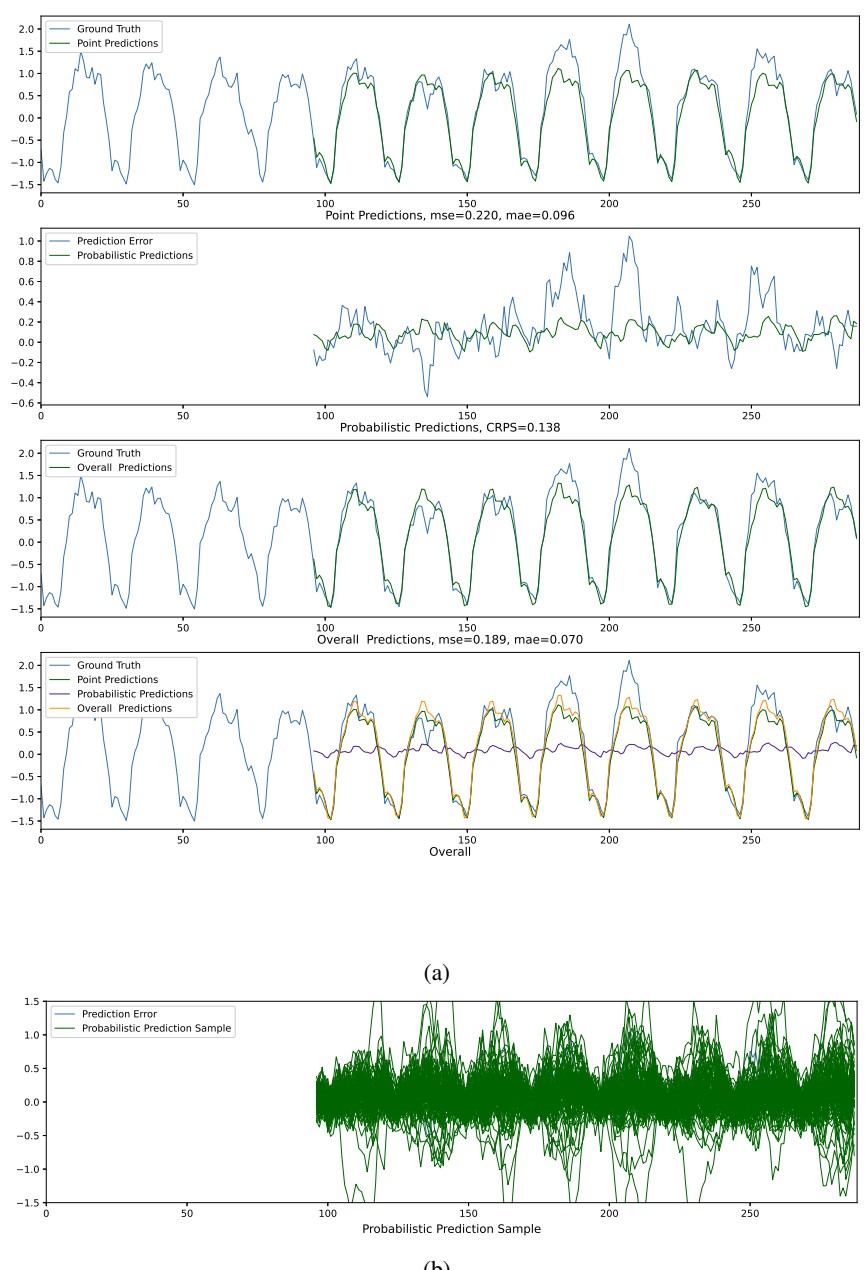

(a)

(b)

Figure 10: Visualization of the Predictions of Eletricity ($140^{th}$ dimension). This case illustrates a scenario with relatively low epistemic uncertainty. Probabilistic prediction: the mean of the diffusion model's samples.

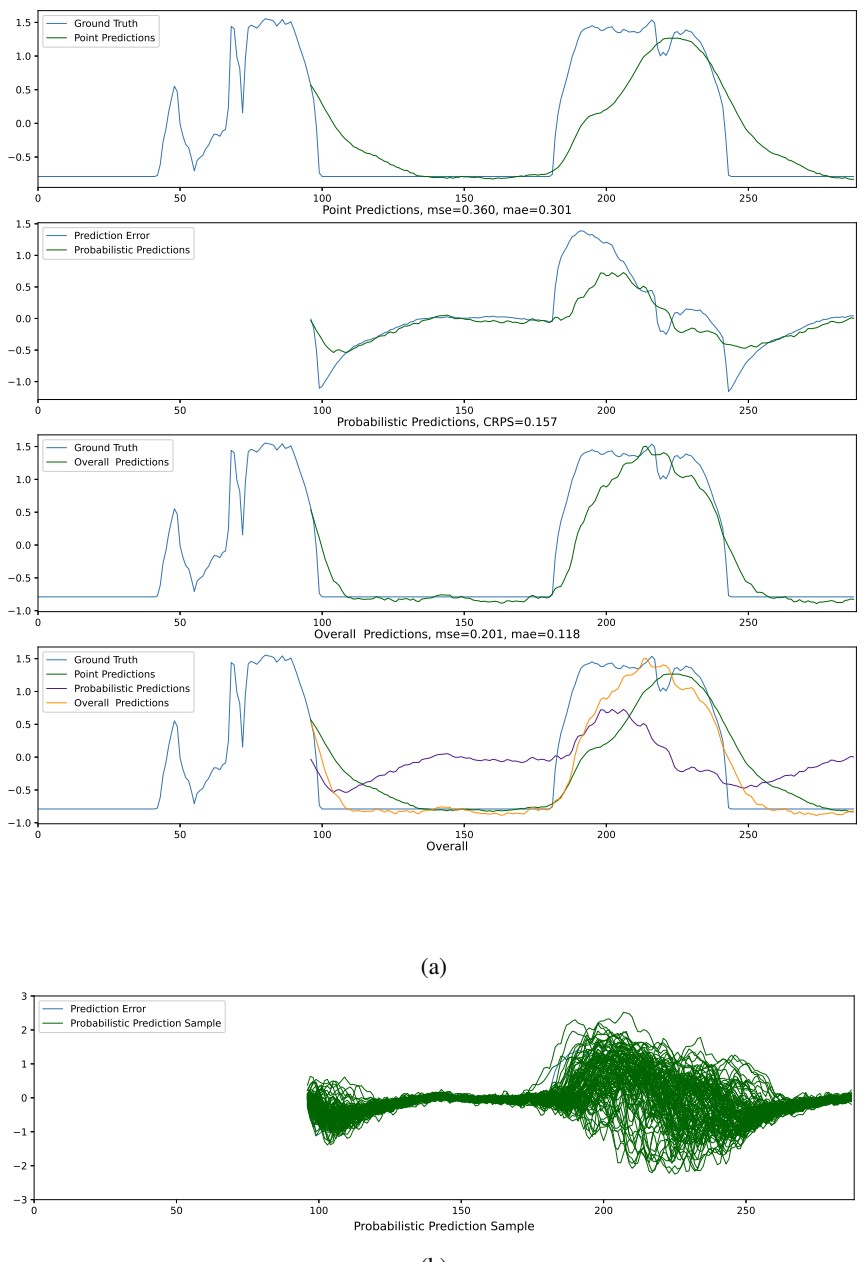

(a)

(b)

Figure 11: Visualization of the Predictions of Solar ($30^{th}$ dimension). This case illustrates a scenario with relatively high epistemic uncertainty. Probabilistic prediction: the mean of the diffusion model's samples.

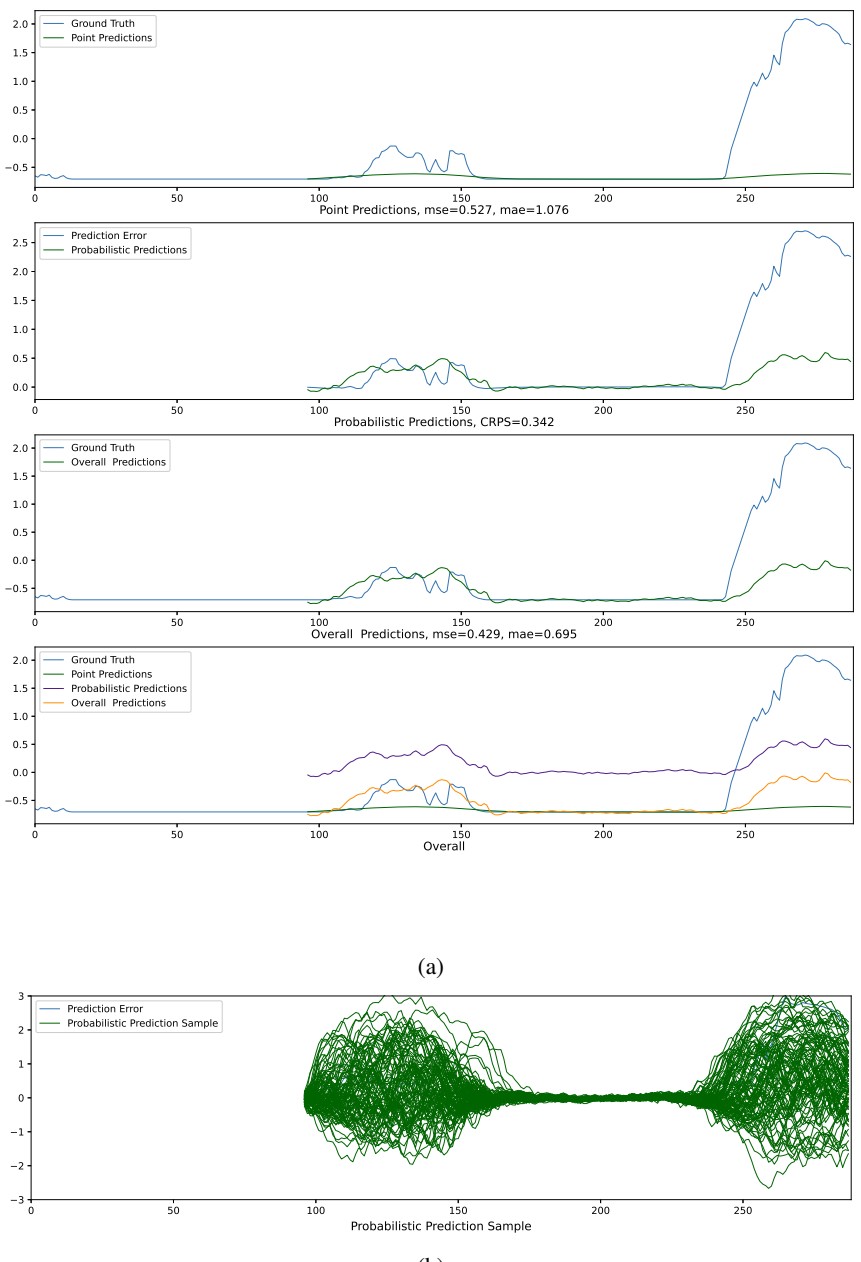

(a)

(b)

Figure 12: Visualization of the Predictions of Solar ($50^{th}$ dimension). This case illustrates a scenario with relatively high epistemic uncertainty. Probabilistic prediction: the mean of the diffusion model's samples.

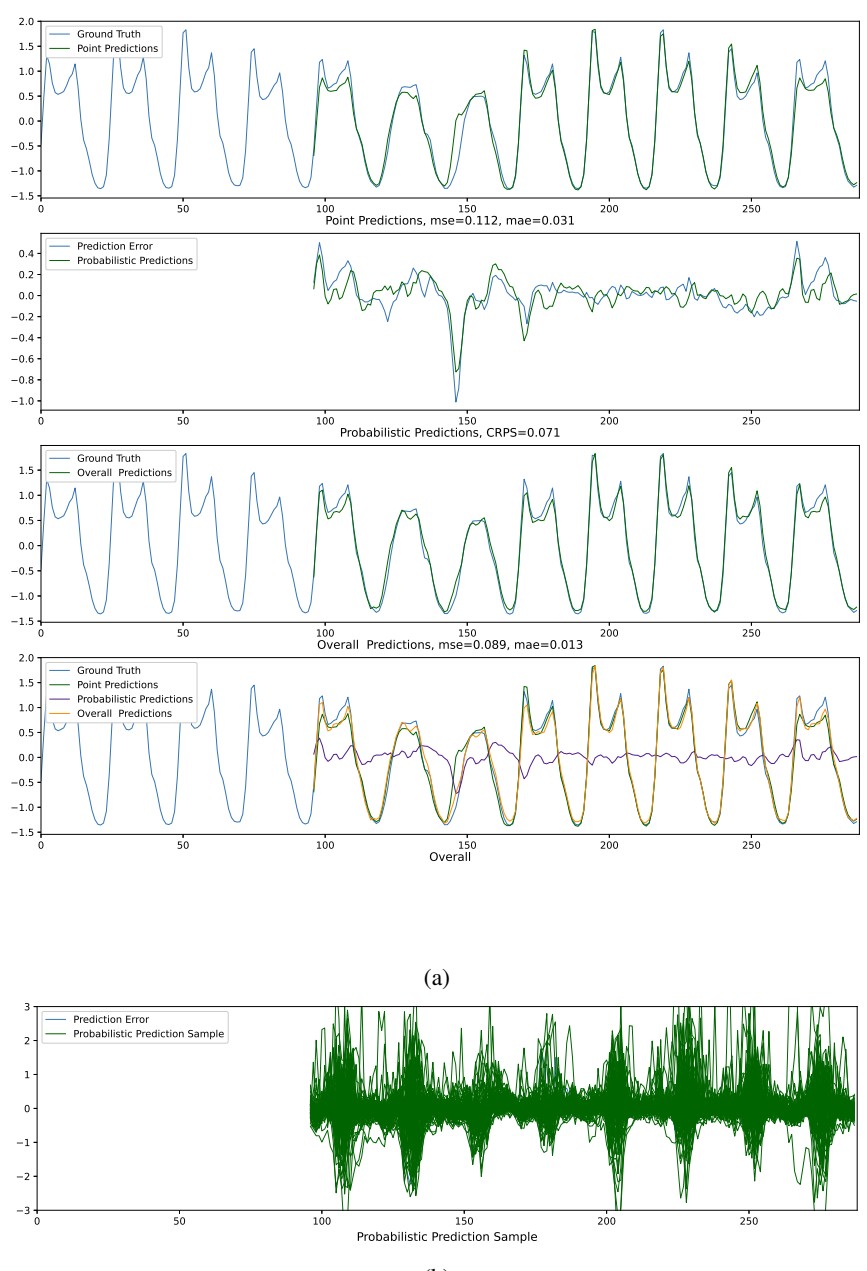

(a)

(b)

Figure 13: Visualization of the Predictions of Traffic ($20^{th}$ dimension). Probabilistic prediction: the mean of the diffusion model's samples.

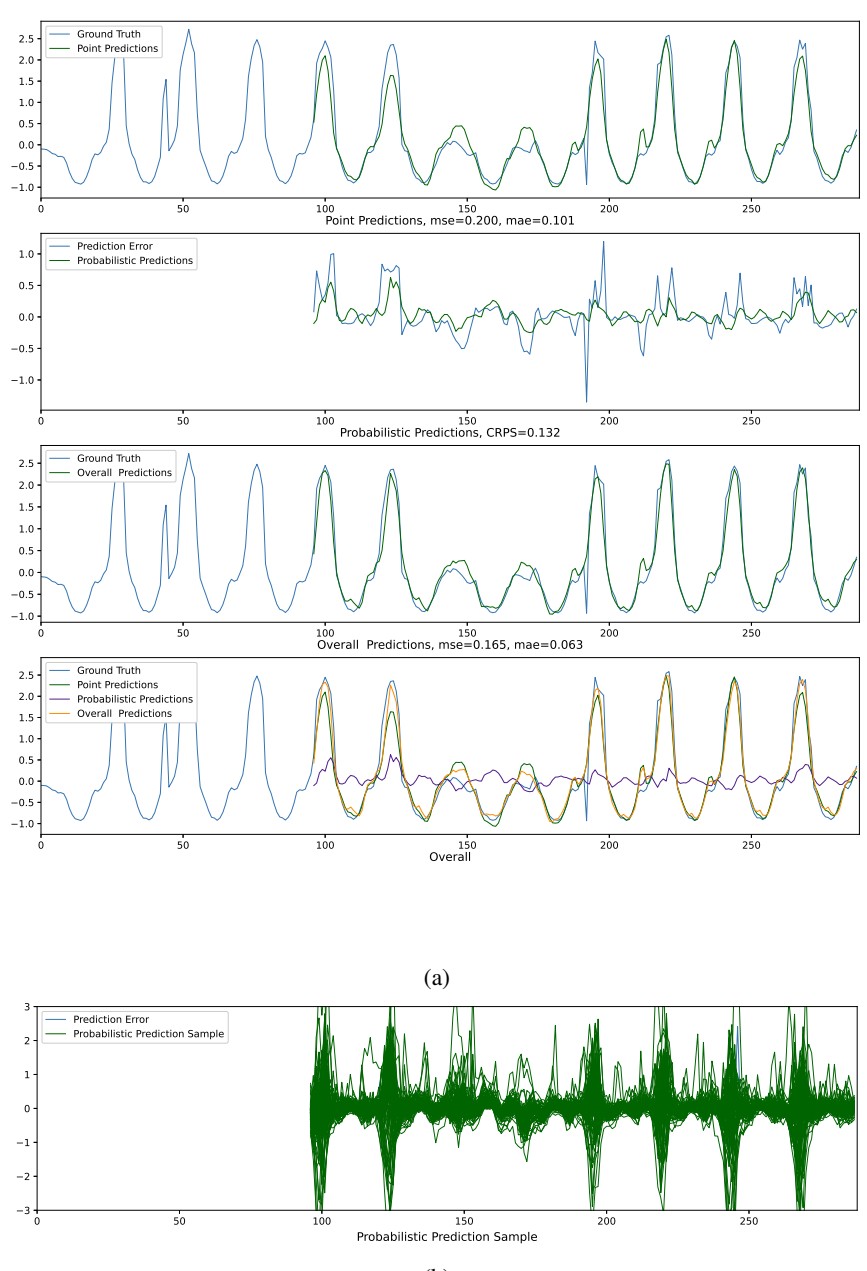

(a)

(b)

Figure 14: Visualization of the Predictions of Traffic ($77^{th}$ dimension). Probabilistic prediction: the mean of the diffusion model's samples.

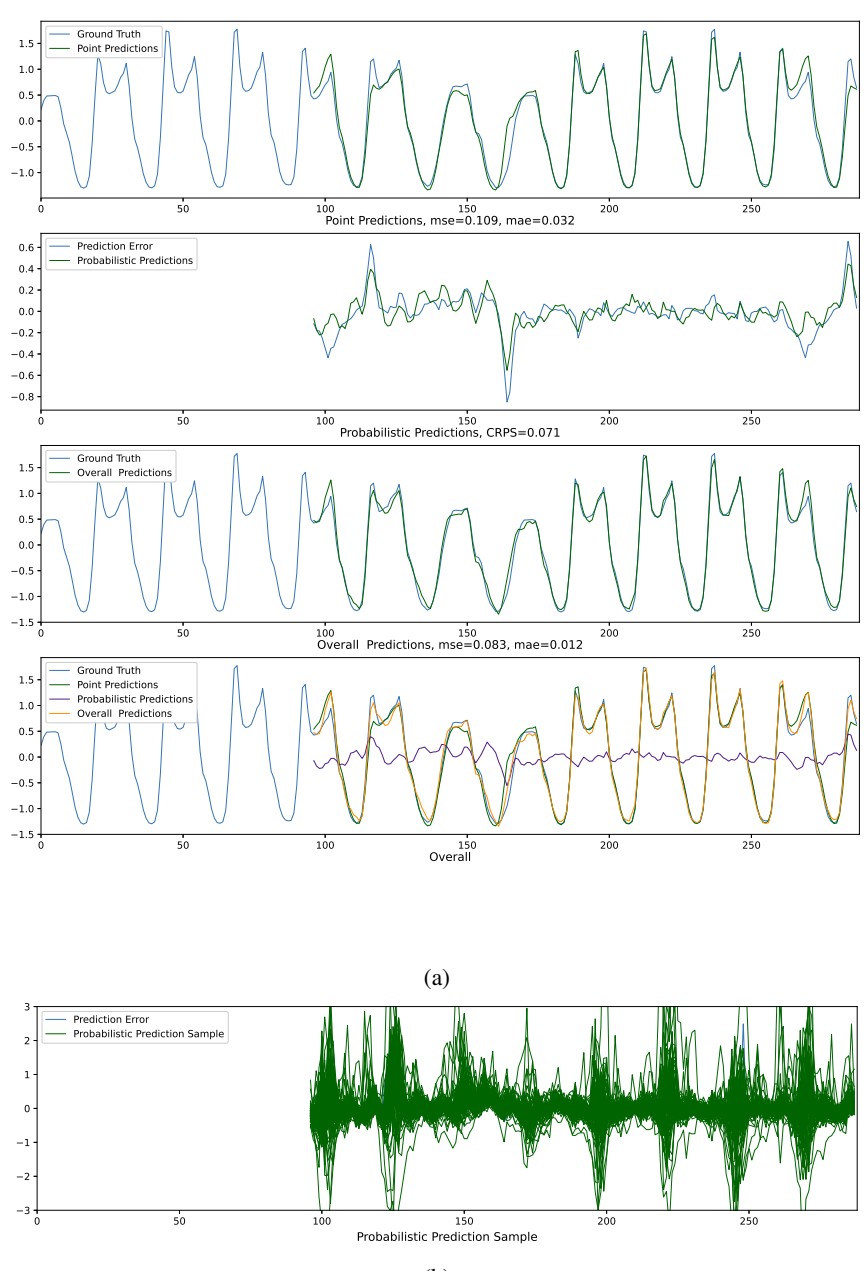

(a)

(b)

Figure 15: Visualization of the Predictions of Traffic ($300^{th}$ dimension). Probabilistic prediction: the mean of the diffusion model's samples.

