# OpenReview forum: "Diffusion-based Decoupled Deterministic and Uncertain Framework for Probabilistic Multivariate Time Series Forecasting"
_ICLR.cc/2025/Conference — ICLR 2025 Poster_

### Official Review · Reviewer_pH5Z · 2024-10-21

**Soundness:** 2
**Presentation:** 3
**Contribution:** 2
**Rating:** 6
**Confidence:** 3

**Summary:**

This paper introduces D3U for probabilistic multi-variable time series (MTS) forecasting, integrating non-probabilistic forecasting with conditional diffusion generation. This enables the framework to produce both accurate point predictions and probabilistic forecasts. Experiments conducted on six real-world datasets demonstrate the superior performance of D3U.

**Strengths:**

1. The D3U framework innovatively combines a point forecasting model to handle high-certainty components within the time series non-probabilistically, while employing a diffusion model to capture the probabilistic distribution of high-uncertainty components. This approach represents an advancement over previous diffusion models that typically model the entire series.

2.D3U is designed as a plug-and-play framework that can be seamlessly integrated with existing point forecasting models. Additionally, the paper introduces a novel patch-based denoising network, PatchDN.

3.The overall organization of the paper is clear, and the paper is understandable in most parts.

**Weaknesses:**

1.My primary concern is the effectiveness of the diffusion model within the framework. I have detailed my questions regarding the actual utility of the diffusion model in predictions in Question 1.

2.The diffusion-based time series models included for comparison are somewhat limited, excluding several advanced diffusion-based models developed in the last two years.

3.As indicated in Table 1, the proposed model performs worse than state-of-the-art point forecasting models in most settings.

4.The model's complexity is notably higher compared to traditional point-to-point models, which might restrict its applicability in practical scenarios.

**Questions:**

1.What are the contribution proportions of the point forecasting model and the diffusion model to the final prediction? It is recommended that the authors provide a table of the respective proportions of point forecasting and probabilistic forecasting for different datasets and include visualizations of some examples. This would help readers understand the roles of the point forecasting model and the diffusion model within D3U and confirm the effectiveness of the diffusion model.

2.The results presented in Figure 1 are interesting. Have the authors explored a similar approach to decompose the series and then used a point forecasting model to predict the deterministic part, while employing a diffusion model to predict the residual obtained from the decomposition?

3.Are the weights of the pre-trained point forecasting model fixed in the D3U framework? Have the authors experimented with fine-tuning the point forecasting model simultaneously while training the diffusion model, or considered creating an end-to-end architecture for the entire model?

---

> ### Author Response · Authors · 2024-11-19
> **Question1 & Weakness1**
>
> Thank you for your valuable comments and suggestions. We combine the results of point forecasting and probabilistic forecasting in a 1:1 ratio across all datasets. In the D3U framework, the diffusion model is used to model the distribution of the high-uncertainty components in time series data that are difficult for the point forecasting model to capture. As explained in Section 3 (page 5, lines 233-243), we use the prediction error $y - \hat{y}$ from the point forecasting model as the input to the diffusion model. Thus, in our implementation, the diffusion model's task is to model the distribution of the point forecasting model's prediction error. Consequently, the final prediction of the overall model is obtained by directly adding the diffusion model's sampled output $\hat{y}_g$ to the point forecasting model's prediction, in a 1:1 ratio.
>
> In light of your suggestions, we have included visualizations of prediction results evolving through the diffusion steps in the reverse process to help readers better understand the respective contributions of point forecasting and probabilistic forecasting to the final prediction results. We have provided these visualizations in Appendix F (page 18, Subsection F.2) of the revised version of the paper.

---

> ### Author Response · Authors · 2024-11-19
> **Question 2**
>
> Thank you for your valuable comments and suggestions. Yes, we have tested the configuration you mentioned, where $x_T + x_S$ is used as the input to the point forecasting model, and the residual component obtained from the decomposition, $x_R := x - x_T - x_S$, is used as the input to the diffusion model (referred to as $\mathrm{D^3U}$-DCP). The experimental results indicate that this configuration yields results that are slightly worse than those obtained with the setup we adopted in the paper. The experimental results, now integrated into the updated version, are presented as follows ("**Bold**/*italic* denotes the **best**/*second-best* results."):
> |      Training Mode    |           |   ETTm1   |           |           |   ETTm2   |           |           |  Weather  |           |           | Solar-Energy |           |           | Electricity |           |           |  Traffic  |           |
> |:---------------------:|:---------:|:---------:|:---------:|:---------:|:---------:|:---------:|:---------:|:---------:|:---------:|:---------:|:------------:|-----------|:---------:|:-----------:|-----------|:---------:|:---------:|-----------|
> |                       |    MSE    |    MAE    |    CRPS   |    MSE    |    MAE    |    CRPS   |    MSE    |    MAE    |    CRPS   |    MSE    |      MAE     |    CRPS   |    MSE    |     MAE     |    CRPS   |    MSE    |    MAE    |    CRPS   |
> | $\mathrm{D^3U}$-DCP | 0.395     | 0.405     | 0.330     | 0.251     | 0.312     | 0.265     | 0.228     | 0.268     | 0.240     | 0.294     | 0.324        | 0.272     | 0.275     | 0.377       | 0.280     | 0.790     | 0.510     | 0.391     |
> | $\mathrm{D^3U}$(Vanilla) | **0.363** | **0.386** | **0.285** | **0.241** | **0.302** | **0.243** | **0.222** | **0.264** | **0.207** | **0.237** | **0.270**    | **0.186** | **0.179** | **0.267**   | **0.202** | **0.468** | **0.299** | **0.232** |
>
> We think the reason for this result is that the trend/seasonal/residual decomposition (DCP) provides a coarse approach to separating the high-certainty and high-uncertainty components of time series data. As shown in the experimental results of Table 9 (Section B.2), the effectiveness of DCP is highly dependent on the choice of hyperparameters $k$ and $f$. Therefore, we do not consider DCP a reliable approach for extracting high-certainty or high-uncertainty components in all cases. In the setup we adopt in the paper, the undecomposed data $x$ is used as the input to the point forecasting model, and the prediction error $y - \hat{y}$ from the point forecasting model is used as the input to the diffusion model. As analyzed in Section 2.1, the prediction error of the point forecasting model primarily consists of two types of uncertainty: aleatoric uncertainty and epistemic uncertainty. Thus, the prediction error is a reasonable approximation of the high-uncertainty component in the time series data that is difficult for the point forecasting model to capture.

---

> ### Author Response · Authors · 2024-11-19
> **Question3**
>
> Thank you for your valuable comments and suggestions. In the D3U framework, the parameters of the pre-trained point forecasting model are frozen, which helps reduce the overall training time of the model. Based on your suggestions, we have experimented with (1) fine-tuning the point forecasting model while training the diffusion model and (2) training the entire model from scratch. The experimental results, now integrated into the updated version, are presented as follows ("**Bold**/*italic* denotes the **best**/*second-best* results."):
>
> |          Training Mode        |           |   ETTm1   |           |           |   ETTm2   |           |           |  Weather  |           |           | Solar-Energy |           |           | Electricity |           |           |  Traffic  |           |
> |:-----------------------------:|:---------:|:---------:|:---------:|:---------:|:---------:|:---------:|:---------:|:---------:|:---------:|:---------:|:------------:|:---------:|:---------:|:-----------:|:---------:|:---------:|:---------:|:---------:|
> |                               |    MSE    |    MAE    |    CRPS   |    MSE    |    MAE    |    CRPS   |    MSE    |    MAE    |    CRPS   |    MSE    |      MAE     |    CRPS   |    MSE    |     MAE     |    CRPS   |    MSE    |    MAE    |    CRPS   |
> |   $\mathrm{D^3U}$(Finetune)   |   0.368   |   0.388   |   0.286   |   0.253   |   0.322   |   0.247   | **0.222** |  _0.271_  | **0.202** | **0.217** |   **0.259**  | **0.169** | **0.174** |  **0.266**  | **0.197** |   0.535   | **0.292** | **0.224** |
> | $\mathrm{D^3U}$(From Scratch) |  _0.364_  | **0.386** | **0.284** |  _0.248_  |  _0.317_  | **0.243** |   0.230   |   0.284   |   0.210   |  _0.220_  |    _0.262_   |  _0.172_  |  _0.177_  |  **0.266**  |  _0.199_  |  _0.516_  |   0.304   |   0.233   |
> | $\mathrm{D^3U}$(Vanilla) | **0.363** | **0.386** |  _0.285_  | **0.241** | **0.302** | **0.243** | **0.222** | **0.264** |  _0.207_  |   0.237   |     0.270    |   0.186   |   0.179   |   _0.267_   |   0.202   | **0.468** |  _0.299_  |  _0.232_  |
>
> The results indicate that, on most datasets, the performance of fine-tuning and training from scratch are not significantly different from the original D3U. However, the training approach using the frozen pre-trained model yields the minimum number of training parameters and the shortest training time. The details of the training times for the different training modes are provided below:
> |    Training Mode |                 ETTm1                 |                         |                 ETTm2                 |                         |                Weather                |                         |              Solar-Energy             |                         |              Electricity              |                         |                Traffic                |                         |
> |:----------------:|:-------------------------------------:|:-----------------------:|:-------------------------------------:|:-----------------------:|:-------------------------------------:|:-----------------------:|:-------------------------------------:|:-----------------------:|:-------------------------------------:|:-----------------------:|:-------------------------------------:|:-----------------------:|
> |                  | Trainable  Parameters ($\times 10^6$) | Training time (s/batch) | Trainable  Parameters ($\times 10^6$) | Training time (s/batch) | Trainable  Parameters ($\times 10^6$) | Training time (s/batch) | Trainable  Parameters ($\times 10^6$) | Training time (s/batch) | Trainable  Parameters ($\times 10^6$) | Training time (s/batch) | Trainable  Parameters ($\times 10^6$) | Training time (s/batch) |
> |   $\mathrm{D^3U}$(Finetune)  |                 7.380                 |          7.471          |                 7.380                 |          7.493          |                 4.825                 |          18.106         |                 3.504                 |         105.355         |                 10.249                |         306.249         |                 6.467                 |         341.098         |
> | $\mathrm{D^3U}$(From Scrach) |                 7.380                 |          7.552          |                 7.380                 |          7.471          |                 4.825                 |          18.050         |                 3.504                 |         105.139         |                 10.249                |         306.112         |                 6.467                 |         340.347         |
> |  $\mathrm{D^3U}$(Vanilla) |               **1.659**               |        **4.567**        |               **1.659**               |        **4.621**        |               **3.024**               |        **13.712**       |               **1.322**               |        **76.627**       |               **2.166**               |       **198.587**       |               **3.495**               |       **316.330**       |

---

> ### Author Response · Authors · 2024-11-19
> **Weaknesses 3-4**
>
> **Weakness 3**
>
> Thank you for your comment. The core idea of the D3U framework is to decouple the deterministic and uncertain components in time series data without imposing specific methods for obtaining these components. In all experiments conducted in the paper, unless otherwise stated, the point forecasting model used in the D3U framework is SparseVQ. (In light of your feedback, we have emphasized this in the experimental section of the paper.) Therefore, in Table 1, the forecasting performance of D3U is comparable to that of SparseVQ. The performance gap between D3U and state-of-the-art point forecasting models (such as iTransformer) in this table is primarily due to the difference in point forecasting capabilities between SparseVQ and these models. Hence, a more objective evaluation in Table 1 would be to compare D3U with SparseVQ's point forecasting performance. As shown, D3U demonstrates point forecasting capabilities that are either comparable to or better than SparseVQ across all datasets. Additionally, as shown in Table 6, when using other point forecasting models, the D3U framework can maintain their original point forecasting performance while enhancing their probabilistic forecasting ability.
>
> **Weakness 4**
>
> Thank you for your comment. Thanks to the design of PatchDN, the computational complexity of the D3U framework, when using PatchDN as the denoising network for the diffusion model, is $O\left(\frac{CH^2}{P^2}\right)$, where $C$ represents the number of variables in the MTS data, $H$ is the length of the input data, and $P$ is the patch length. Therefore, when the computational complexity of the point forecasting model in D3U exceeds $O\left(\frac{CH^2}{P^2}\right)$, the overall computational complexity is primarily determined by the point forecasting model. On the other hand, when the complexity of the point forecasting model is lower, the overall complexity remains $O\left(\frac{CH^2}{P^2}\right)$, which is comparable to the computational complexity of many state-of-the-art point forecasting models $\underline{[1,2,3]}$.
>
> [1] Nie Y, Nguyen N H, Sinthong P, et al. A time series is worth 64 words: Long-term forecasting with transformers[J]. arXiv preprint arXiv:2211.14730, 2022.
>
> [2] Zhang Y, Yan J. Crossformer: Transformer utilizing cross-dimension dependency for multivariate time series forecasting[C]//The eleventh international conference on learning representations. 2023.
>
> [3] Zhang Z, Meng L, Gu Y. SageFormer: Series-Aware Framework for Long-Term Multivariate Time Series Forecasting[J]. IEEE Internet of Things Journal, 2024.

---

> ### Author Response · Authors · 2024-11-21
> **Weakness 2**
>
> **Weakness 2**
>
> Thank you for your comments. Since the main focus of this paper is long-term time series forecasting, we specifically searched for diffusion-based models with open-source code that address long-term forecasting challenges, including TimeDiff (2023) and TMDM (2024). The remaining diffusion-based models primarily tackle short-term forecasting tasks. Since short-term forecasting models are designed specifically for short-term data, they often fail to capture certain characteristics of long-term time series data. Consequently, their performance tends to degrade when applied to long-term forecasting tasks. Based on your suggestion, we have added two highly-cited models, TimeGrad (2021) $\underline{[1]}$ and CSDI (2021) $\underline{[2]}$, as additional baseline models. These two models have also been utilized as baselines in the experimental comparisons of TimeDiff and TMDM. The experimental results, now integrated into the updated version, are presented as follows ("**Bold**/*italic* denotes the **best**/*second-best* results."):
>
> Table 1: Performance comparison on six real-world datasets based on MSE and MAE in probabilistic forecasting models.
> |     Dataset    |      ETTm1      |      ETTm2      |     Weather     |   Solar-Energy  |   Electricity   |     Traffic     |
> |:--------------:|:---------------:|:---------------:|:---------------:|:---------------:|:---------------:|:---------------:|
> |     Method     |     MSE MAE     |     MSE MAE     |     MSE MAE     |     MSE MAE     |     MSE MAE     |     MSE MAE     |
> | TimeGrad(2021) |   1.716 1.057   | 1.385 0.732     | 0.885 0.551     | 1.211 1.004     | 0.645 0.723     | 0.932 0.807     |
> |   CSDI(2021)   | 0.867 0.690     | 1.291 0.576     | 0.842 0.523     | 0.848 0.818     |  0.553 0.795    | 0.921 0.678     |
> | TimeDiff(2023) |   0.796 0.577   |  _0.284 0.342_  |   0.277 0.331   |   1.169 0.936   |   0.730 0.690   |   1.465 0.851   |
> |   TMDM(2024)   |  _0.607 0.558_  |   0.524 0.493   |  _0.244 0.286_  |  _0.295 0.317_  |  _0.222 0.329_  |  _0.721 0.411_  |
> |      ours      | **0.363 0.386** | **0.241 0.302** | **0.222 0.264** | **0.237 0.270** | **0.179 0.267** | **0.468 0.299** |
>
> Table 2: Performance comparisons on six real-world datasets regarding CRPS and CRPS_sum in probabilistic forecasting models.
> |     Dataset    |      ETTm1      |      ETTm2      |     Weather     |   Solar-Energy  |    Electricity    |      Traffic      |
> |:--------------:|:---------------:|:---------------:|:---------------:|:---------------:|:-----------------:|:-----------------:|
> |     Method     |  CRPS CRPS_sum  |  CRPS CRPS_sum  |  CRPS CRPS_sum  |  CRPS CRPS_sum  |   CRPS CRPS_sum   |   CRPS CRPS_sum   |
> | TimeGrad(2021) |   0.665 0.996   |   0.785 1.051   |   0.482 0.503   |   0.783 1.167   |    0.503 1.452    |    0.657 1.683    |
> |   CSDI(2021)   |   0.773 0.852   |   0.625 0.782   |   0.508 0.465   |   0.649 0.681   |    0.465 0.823    |    0.612 1.275    |
> | TimeDiff(2023) |   0.454 0.846   |  _0.316 0.180_  |   0.293 0.400   |   0.900 1.164   |    0.475 0.594    |    0.671 0.823    |
> |   TMDM(2024)   |  _0.429 0.633_  |   0.380 0.226   |  _0.226 0.292_  |  _0.375 0.267_  | _0.446_ **0.137** | _0.552_ **0.179** |
> |      ours      | **0.285 0.574** | **0.243 0.141** | **0.207 0.141** | **0.186 0.266** | **0.202** _0.160_ | **0.232** _0.186_ |
>
> The results demonstrate that D3U outperforms the other four models, exhibiting superior point prediction accuracy and distribution estimation. D3U's remarkable performance stems from two key factors:
>
> 1.D3U decouples the deterministic and uncertain components in multivariate time series (MTS) data. Within the D3U framework, the diffusion model focuses on capturing the high-uncertainty components of the data, while the information related to the more deterministic components is provided by the point forecasting model. This design not only preserves the point forecasting ability of D3U but also reduces the learning difficulty for the diffusion model.
>
> 2.The proposed PatchDN enhances the diffusion model's ability to capture both the guidance information from the point forecasting model and the temporal features of time series data.
>
> [1] Rasul K, Seward C, Schuster I, et al. Autoregressive denoising diffusion models for multivariate probabilistic time series forecasting[C]//International Conference on Machine Learning. PMLR, 2021: 8857-8868.
>
> [2] Tashiro Y, Song J, Song Y, et al. Csdi: Conditional score-based diffusion models for probabilistic time series imputation[J]. Advances in Neural Information Processing Systems, 2021, 34: 24804-24816.

---

> ### Author Response · Authors · 2024-11-22
>
> Dear Reviewer pH5Z,
>
> We appreciate it if you could let us know whether our responses are able to address your concerns. We're happy to address any further concerns. Thank you,
>
> Best wishes.
>
> Paper7497 Authors

---

> > ### Comment · Reviewer_pH5Z · 2024-11-25
> >
> > Thank you for your response. Regarding Question1 & Weakness1, I still have some concerns. The visualizations you provided are not exactly what I was looking for. The final prediction of D3U is determined by the combination of point prediction and probabilistic prediction. I would like to visualize the point prediction and probabilistic prediction separately, and if the model's predictions become closer to the ground truth after incorporating the probabilistic prediction. Could you please demonstrate some visualizations that include point predictions, probabilistic predictions, overall predictions, and the ground truth on the same plot?
> >
> > My concerns arise from the observation that, according to Table 1, the results of D3U and SparseVQ are quite similar. This similarity leads me to question whether the incorporation of probabilistic predictions and the utilization of diffusion models merely introduce randomness without substantially enhancing the model's predictive accuracy. Although integrating uncertainty is undoubtedly a contribution, I would still like to see evidence that the introduction of the diffusion model enhances the model's predictive accuracy.

---

> > > ### Author Response · Authors · 2024-11-25
> > >
> > > Thank you for your valuable comments. We are in the process of adding new visualizations based on your suggestions. The revised version of the paper, including the new visualization content, will be uploaded soon, and we will notify you once it is available.

---

> ### Author Response · Authors · 2024-11-25
> **Question1 & Weakness1，new visualizations and analysis**
>
> Thank you for your valuable comment. In light of your feedback, we have added new visualizations (Section F.3, APPENDIX) that present point prediction results, probabilistic prediction results, overall prediction results, and corresponding ground truth on the same plot. To address potential concerns about overlapping curves obscuring details, we also provide separate visualizations for each of the aforementioned prediction components. Additionally, we include further analyses of the diffusion model's outputs to better illustrate its capability in modeling data uncertainty. The new visualization content shows that in the D3U framework, diffusion models can effectively improve the overall predictive accuracy of the model by learning the time-dependent data distribution of high-uncertainty components in time-series data.We hope this response addresses your concerns effectively.

---

> > ### Comment · Reviewer_pH5Z · 2024-11-26
> >
> > Thank you for your efforts. Based on the new visualization samples provided by the authors, it appears that the diffusion model (probabilistic prediction) does not predict the prediction error well. Consequently, I still have some reservations about the claim that the diffusion model can enhance predictive performance. Nonetheless, the addition of uncertainty into predictions is a clear contribution, and I appreciate the authors' efforts to address my concerns. I have decided to increase my score to 6. I hope that the authors will conduct further explorations in future work to improve the diffusion model's ability to predict prediction errors more accurately and to better leverage its potential to enhance predictive performance.

---

> ### Author Response · Authors · 2024-11-26
>
> Once again, we would like to express our sincere appreciation for the reviewer's efforts and the time dedicated to providing valuable comments and suggestions for our work.
>
> The D3U framework introduces a novel approach for long-term MTS probabilistic forecasting based on uncertainty decoupling. We hope D3U can provide a new perspective for the field of long-term MTS forecasting, particularly in probabilistic forecasting. As you mentioned, there is still significant room for improvement in modeling uncertainty, such as further exploration of aleatoric and epistemic uncertainty in time-series data (Section 4.3). However, the case of high epistemic uncertainty illustrated in Section 4.3 and Figure 3 represents an extreme scenario. In most cases, as shown in the newly added visualizations (Fig. 11 and Fig. 12), even when faced with relatively high epistemic uncertainty—where the point forecasting model's results significantly deviate from the ground truth—the diffusion model can still effectively capture the distribution of prediction errors and improve the overall prediction accuracy.
>
> In Fig. 9, Fig. 10, Fig. 11, and Fig. 12, we used the mean of 100 samples generated by the diffusion model to provide a more intuitive comparison between probabilistic predictions and the ground truth of prediction errors. However, it is important to note that D3U's diffusion model is designed for probabilistic forecasting, with the primary goal of modeling the underlying distribution of prediction errors. Therefore, a more appropriate way to compare the output of probabilistic forecasting with ground truth is to use probabilistic evaluation metrics, such as the Continuous Ranked Probability Score (CRPS), which directly compare the distribution predicted by the model with individual ground truth values, rather than assessing the difference between the mean of the samples and the ground truth values with point prediction metrics like MSE or MAE.
>
> In the visualizations, the average CRPS for samples from the Electricity (Fig. 9, Fig. 10) and Solar-Energy (Fig. 11, Fig. 12) datasets are 0.125 and 0.249 (the CRPS for each case has been added to the corresponding figure), respectively, outperforming all baselines on the corresponding datasets as reported in Table 2.
>
> Additionally, in the latest revised version, we have included $\underline{\text{extra visualization cases}}$ (Fig.13, Fig.14 and Fig.15) from the Traffic dataset. These cases aim to further demonstrate the effectiveness of the diffusion model within the D3U framework in modeling the distribution of prediction errors.  If this response helps to address your concerns and clarify the points raised, we kindly hope you to consider raising our rating. We deeply value your constructive feedback and have made every effort to incorporate your suggestions to enhance the clarity and quality of our work. Thank you for your time and thoughtful review.

---

### Official Review · Reviewer_igxL · 2024-10-27

**Soundness:** 2
**Presentation:** 3
**Contribution:** 2
**Rating:** 6
**Confidence:** 4

**Summary:**

The D3U framework is introduced for probabilistic multivariate time series (MTS) forecasting. D3U combines a point forecasting model for high-certainty components with a conditional diffusion model to capture high-uncertainty components. D3U achieves excellent performance across six real-world datasets and includes comprehensive ablation studies.

**Strengths:**

Overall, the paper is well-written and easy to follow. The details in each section are well-explained, and the experiments are thorough.

**Weaknesses:**

The main concern is the originality of the paper. The concept is straightforward: decomposing multivariate time series data into deterministic and uncertain components, using a point forecasting model for the high-certainty component and a conditional diffusion model for the high-uncertainty component. Most of the diffusion design relies on previous work; for example, SparseVQ (Zhao et al., 2024) is used as the conditioning network, and the conditional DDPM is almost the same as the original. The only apparent innovation is the design of the Patch Denoising Network, which itself is not overly complex. Overall, this work feels more like a stitching together of existing methods applied to time series data, with insufficient original contributions.

**Questions:**

The proposed method requires decomposing the time series data. As shown in Appendix B, decomposing the time series data requires some prior knowledge or specific hyperparameter settings. A natural question arises: when deploying the trained model in a new scenario, you often don’t know how the training data was decomposed. So, how would you decompose the data in the new scenario? Additionally, if the characteristics of the deterministic and uncertain components in the time series data differ between the new scenario and the training data, how can you ensure the model performs well?

---

> ### Author Response · Authors · 2024-11-19
> **Question**
>
> Thank you for your valuable comment. In our paper, only the two case studies in Section 1 and the experiment in Section B.2 of the Appendix use the series decomposition technology (referred to as DCP) to acquire the trend/seasonal components and residual components of the time series. The use of DCP in Section 1's case studies assists our analysis of the point forecasting model's ability to capture different components of time series data with varying levels of uncertainty. These two case studies indicate that point forecasting models have better modeling capability for components with higher determinacy than those with higher uncertainty. This observation inspires us to leverage the complementary strengths of point forecasting and diffusion models in time series data modeling. The experiment on two hyperparameters (kernel $k$ and Fourier factor $f$) of DCP in Appendix B.2 aims to demonstrate how different hyperparameter settings affect the final performance when using the high-certainty component $x_T + x_S$ ($x_T$ and $x_S$ represent the trend and seasonal components, respectively) as the input for point forecasting models. The optimal parameter settings vary across different point forecasting models. This experiment provides foundational guidance for practitioners considering using $x_T + x_S$ as input within the D3U framework. Additionally, given that some point forecasting models, such as DLinear $\underline{[1]}$, inherently integrate DCP in their design, we retain the option to use $x_T + x_S$ as model input.
>
> In the rest of the paper, we do not use DCP to model high-certainty or high-uncertainty components of time series data. As shown in the experimental results of Table 9 (Section B.2), the effectiveness of DCP is highly dependent on the choice of hyperparameters $k$ and $f$. Therefore, we also do not consider DCP a reliable approach for extracting high-certainty or high-uncertainty components in all cases.
>
> We have opted for a more straightforward and more effective approach: using the entire data $x$ as input for the point forecasting model and treating the model's prediction error as the high-uncertainty component, which is then used as input for the generative model. As analyzed in Section 2.1, the prediction error of the point forecasting model primarily consists of two types of uncertainty: aleatoric uncertainty and epistemic uncertainty. Thus, the prediction error is a reasonable approximation of the high-uncertainty component in the time series data that is difficult for the point forecasting model to capture.
>
> [1] Zeng A, Chen M, Zhang L, et al. Are transformers effective for time series forecasting?[C]//Proceedings of the AAAI conference on artificial intelligence. 2023, 37(9): 11121-11128.

---

> ### Author Response · Authors · 2024-11-19
> **Weaknesses**
>
> Thank you for your valuable comment. Based on the two case studies in Section 1, we observed that point forecasting models are more adept at capturing components with lower uncertainty while struggling with components characterized by higher uncertainty. Inspired by this, we propose a novel complementary modeling approach that combines point forecasting models and probabilistic forecasting models from the perspective of decoupling the deterministic and uncertain components of time series data. Based on the idea of complementary modeling, we propose the D3U framework. D3U leverages a pre-trained point forecasting model to learn the high-certainty components and injects valuable representations of these components as conditional information into the reverse process of a conditional DDPM, aiding it in performing probabilistic forecasting on the high-uncertainty components.
>
> PatchDN is designed to enhance the representation capability of the conditional diffusion model for time series data. However, even when it is replaced with an MLP (Table 4, SVQ+MLP), the D3U framework still achieves excellent point forecasting and probabilistic forecasting performance, outperforming the baselines in the Table 2.
>
> Additionally, our method does not rely on a specific point forecasting model. When using the same configuration as TMDM—NSformer $\underline{[1]}$ as the point forecasting model and MLP as the denoising network for the diffusion model—the D3U framework achieves significant performance improvements (Table 3). The simple uncertainty decoupling modeling approach adopted in the D3U framework enhances its interpretability. *To the best of our knowledge, D3U is the first framework to utilize the decoupling of deterministic and uncertain components in long-term multivariate time series (MTS) probabilistic forecasting.*
>
> [1] Liu Y, Wu H, Wang J, et al. Non-stationary transformers: Exploring the stationarity in time series forecasting[J]. Advances in Neural Information Processing Systems, 2022, 35: 9881-9893.

---

> ### Author Response · Authors · 2024-11-22
>
> Dear Reviewer igxL,
>
> We appreciate it if you could let us know whether our responses are able to address your concerns. We're happy to address any further concerns. Thank you,
>
> Best wishes.
>
> Paper7497 Authors

---

> > ### Comment · Reviewer_igxL · 2024-11-26
> >
> > Thanks for the authors' reply. I decide to increase the score.

---

> > > ### Author Response · Authors · 2024-11-26
> > >
> > > Dear Reviewer igxL,
> > >
> > > We sincerely thank you for your careful and detailed reading of our paper and responses.
> > >
> > > Best regards,
> > >
> > > Paper7497 Authors

---

### Official Review · Reviewer_jQCt · 2024-11-02

**Soundness:** 3
**Presentation:** 3
**Contribution:** 2
**Rating:** 6
**Confidence:** 4

**Summary:**

The authors describe a novel framework (D3U) for probabilistic time series forecasting. It combines the point forecasting of seasonal and trend components with a generative sampling model for the residual component.
The point forecasting part consists of a well-known model (SparseVQ) and is used to produce a prediction $\hat y$ of the input time series as well as a latent representation $f_{enc}(x)$ which serves as a condition for the generative part.
The generative part is a DDPM which shall produce samples of the residual $y-\hat y$, the reverse process is hereby conditioned on the latent $f_enc(x)$ and the conditional expectation in the reverse process is modelled by a novel patch-based denoising network $g_\phi$.
The network $g_\phi$ is comprised of a sine-positional encoder for a strided windows representation of the residuals and transformer encoder blocks. It is conditioned on the latent $f_enc(x)$.
In experiments, the model shows competitive or better performance compared to SOTA models on different datasets. The authors conclude that the architecture can serve as framework to obtain a generative sampling model of improved performance for arbitrary point forecasting, conditioning models.

**Strengths:**

The authors show a general way of extending classical point forecasting models to generative sampling models. The paper contains a comprehensive and well-founded survey of existing approaches and works. The results show improved performance compared to SOTA approaches.

**Weaknesses:**

The general idea to split the prediction of a time series in the prediction of trend/seasonal components and the residual is well-known, as is the usage of generative approaches to model the residual. However, the details of the architecture are new.

**Questions:**

p2: the residual component tends to contain more uncertainty than the trend and seasonal components: this is time-series specific.

p3: contribution 2: is this a contribution? It looks like a second description of D3U and hence an additional paragraph belonging to the first contribution.

p4: In statistics, $\cal F(X)$ is the sigma-algebra generated by $X$. Better use another symbol.

p5: The symbols $x$, … are not introduced.

p5 l.243: $\hat y_g+E(\hat y|x_h)$ should be $\hat y_g+\hat y$

p6 l. 281: check the formula of $\mu_\phi$ (how is it derived? Shouldn't be $a$ instead of $\bar a$ and the $\beta$ outside the square root?).

p6 l. 297: check the formula $N=\dots$.

p6 l. 322: $p^k$ is not introduced.

p7: check the formulas (12) and (13) for typos ($r^k_{Pembed}$ is used multiple times)

p10: section 4.3.: there were better results obtained when SparseVQ was replaced by iTransformer. It should be explained why SparseVQ is proposed in the rest of the paper.

---

> ### Author Response · Authors · 2024-11-19
> **Question**
>
> We sincerely thank you for careful reading. In light of your comments, we have revised the text and symbols in the paper to ensure greater clarity and consistency. Due to character limitations in the reply, the responses to these issues need to be provided in batches. Some questions are not addressed in the order you provided, but we will include the corresponding question before each reply.
>
> (i) p2: The relevant descriptions have been clarified to explicitly refer to the time series data: the residual component of time series data tends to contain more uncertainty than the trend and seasonal components.
>
> (ii) p3: contribution 2. Thank you for your valuable suggestion. Our intention in the first contribution is to emphasize the modeling approach of uncertainty decoupling derived from the case studies. This concept leverages the complementary strengths of point forecasting models and probabilistic forecasting models in time series modeling. In the second contribution, we proposed a specific modeling framework, D3U, based on the aforementioned approach. We acknowledge that the original contribution statements have shortcomings in clarity and expression. Based on your suggestion, we have refined the contributions of the paper as follows:
>
> 1. Motivated by the phenomenon observed in case studies that point forecasting models exhibit varying capabilities in modeling the deterministic and uncertain components of time series, we propose a novel complementary modeling approach that combines point forecasting models and probabilistic forecasting models from the perspective of decoupling the deterministic and uncertain components of time series data. Specifically, point forecasting models and probabilistic forecasting models are used to model the high-certainty and high-uncertainty components of time series data, respectively.
>
> 2. We propose $\mathrm{D^3U}$, a long-term MTS probabilistic forecasting framework based on the complementary modeling. $\mathrm{D^3U}$ leverages a pre-trained point forecasting model to learn the high-certainty components and injects valuable representations of these components as conditional information into the reverse process of a conditional DDPM, aiding it in performing probabilistic forecasting on the high-uncertainty components. The final prediction combines the probabilistic forecast with the non-probabilistic forecast from the point forecasting model, enabling $\mathrm{D^3U}$ to achieve accurate data distribution modeling while maintaining strong point forecasting performance. $\mathrm{D^3U}$ is a plug-and-play framework that can be seamlessly integrated with existing point forecasting models and diffusion-based long-term forecasting models, improving both point and probabilistic forecasting capabilities.
>
> 3. Within the $\mathrm{D^3U}$ framework, we design a patch-based denoising network, PatchDN, to enhance the diffusion model’s ability to represent the high-uncertainty components in time series data. Our method demonstrates excellent probabilistic forecasting performance and competitive point forecasting capabilities across six real-world datasets.
>
> (ii) p4: Symbol F(X) has been replaced by F(X).
>
> (iii) p5: Since it is no longer used, we have removed the symbol $x, \dots$ to avoid confusion with $x_h$ (the input of D3U) in terms of notation.
>
> (iv) $\hat{y}_g+\mathbb{E}(\hat{y}|x_h)$ has been replaced by  $\hat{y}_g+\hat{y}$.
>
> (v) p6 l. 297: check the formula N=⋯. Thank you for your comment. We have re-verified this formula. Following the operation in PatchTST $\underline{[1]}$, we pad $S$ repeated numbers of the last value $r^k_L$ to the end of the original sequence $r^k_{1:L}$ before patching. Therefore, the number of patches is $N=(\lfloor\dfrac{H-P}{S}\rfloor+1)+1$
>
> (vi) p6 l. 322: $p^k$ is not introduced. The original declaration of $p^k$ is located far from its usage. We have optimized this by repositioning the declaration to ensure clear and precise notation.
>
> (vii) p7: check the formulas (12) and (13) for typos. Symbol $r_{Pembed}^k$ has been replaced by $r_{PE}^k$ to ensure clarity.
>
> [1] Nie Y, Nguyen N H, Sinthong P, et al. A time series is worth 64 words: Long-term forecasting with transformers[J]. arXiv preprint arXiv:2211.14730, 2022.

---

> > ### Author Response · Authors · 2024-11-19
> > **Question**
> >
> > (viii) p6 l. 281: check the formula of μ_ϕ (how is it derived? Shouldn't be α instead of  α ̅ and the β outside the square root?).
> >
> > Thank you for your suggestion. We have re-verified formula (7) and revised it as follows:
> > \begin{equation}
> > \mu_\phi(r_{1:L}^k,k|c)\dfrac{\sqrt\alpha_k(1-\bar\alpha_{k-1})}{1-\bar\alpha_k}r_{1:L}^k+\dfrac{\sqrt{\bar\alpha_{k-1}}\beta_k}{1-\bar\alpha_k}r_\phi(r^k_{1:L},k|c).
> > \end{equation}
> >
> > This formula first derives the mean of $q(r_{1:L}^{k-1}|r_{1:L}^k,r_{1:L}^0)$ based on the standard DDPM reverse process $\underline{[1]}$, which is given by:
> >
> > \begin{equation}
> > \mu_k(r_{1:L}^k,r_{1:L}^0,k)=\dfrac{\sqrt\alpha_k(1-\bar\alpha_{k-1})}{1-\bar\alpha_k}r_{1:L}^k+\dfrac{\sqrt{\bar\alpha_{k-1}}\beta_k}{1-\bar\alpha_k}r_{1:L}^0.
> > \end{equation}
> >
> > For the definitions of $α^k≔1-β^k$ and  $\bar{α}^k≔\prod_{i=1}^k{α^i}$ , we follow the notation convention used in DDPM $\underline{[1]}$, which is also adopted by many time series papers based on DDPM $\underline{[2, 3]}$. It is worth noting that TMDM uses the opposite notation for these definitions, but this does not affect the correctness of the final formula.
> >
> > In DDPM, the training objective $D_{KL}\left(q(r_{1:L}^{k-1}|r_{1:L}^k)||q(r_{1:L}^k|r_{1:L}^{k-1})\right)$ is transformed as:
> > \begin{equation}
> > L_k=\dfrac{1}{2\sigma^2_k}||\mu_k(r_{1:L}^k,r_{1:L}^0,k)-\mu_\phi(r_{1:L}^k,k)||^2.
> > \end{equation}
> >
> > $\mu_\phi(r_{1:L}^k,k)$ can be defined in two ways $\underline{[4]}$: (i) computed from a noise prediction model; (ii) computed from a data prediction model. We adopt the second way in our paper, that is, $\mu_\phi(r_{1:L}^k,k)$ is computed by reconstructing the input $r_{1:L}^0$:
> > \begin{equation}
> > \mu_\phi(r_{1:L}^k,k)\dfrac{\sqrt\alpha_k(1-\bar\alpha_{k-1})}{1-\bar\alpha_k}r_{1:L}^k+\dfrac{\sqrt{\bar\alpha_{k-1}}\beta_k}{1-\bar\alpha_k}r_\phi(r^k_{1:L},k),
> > \end{equation}
> >
> > where $r_\phi(r^k_{1:L},k)$ serves as the data prediction model. In our paper, $r_\phi(r^k_{1:L},k)$ is implemented as a patch-based denoising network, the proposed PatchDN. When an additional condition input $c$ is available, this can be injected into the backward denoising step as:
> > \begin{equation}
> > \mu_\phi(r_{1:L}^k,k|c)\dfrac{\sqrt\alpha_k(1-\bar\alpha_{k-1})}{1-\bar\alpha_k}r_{1:L}^k+\dfrac{\sqrt{\bar\alpha_{k-1}}\beta_k}{1-\bar\alpha_k}r_\phi(r^k_{1:L},k|c).
> > \end{equation}
> >
> > [1] Ho J, Jain A, Abbeel P. Denoising diffusion probabilistic models[J]. Advances in neural information processing systems, 2020, 33: 6840-6851.
> >
> > [2] Rasul K, Seward C, Schuster I, et al. Autoregressive denoising diffusion models for multivariate probabilistic time series forecasting[C]//International Conference on Machine Learning. PMLR, 2021: 8857-8868.
> >
> > [3] Shen L, Kwok J. Non-autoregressive conditional diffusion models for time series prediction[C]//International Conference on Machine Learning. PMLR, 2023: 31016-31029.
> >
> > [4] Benny Y, Wolf L. Dynamic dual-output diffusion models[C]//Proceedings of the IEEE/CVF Conference on Computer Vision and Pattern Recognition. 2022: 11482-11491.
> >
> > (ix) p10: section 4.3.: there were better results obtained when SparseVQ was replaced by iTransformer. It should be explained why SparseVQ is proposed in the rest of the paper.
> >
> > Thank you for your valuable comment. In the case study presented in Section 1, we analyze the differences in uncertainty across the trend, seasonal, and residual components of time series data by examining the distribution of codebook vectors from the point forecasting model SparseVQ. Therefore, in the subsequent experiments, we consistently used the SparseVQ model as the point forecasting model in the D3U framework. In Section 4.3, when analyzing the impact of epistemic uncertainty and aleatoric uncertainty on the probabilistic forecasting ability of D3U, we replaced SparseVQ with iTransformer, a model with stronger point forecasting capability, to reduce the contribution of epistemic uncertainty in the prediction error. The D3U framework is not limited to a specific point forecasting model; applying the D3U framework to different point forecasting models consistently enhances their probabilistic forecasting performance while preserving their original point forecasting capabilities, as shown in Table 6.

---

> ### Author Response · Authors · 2024-11-19
> **Weakness**
>
> Thank you for your comment. The core idea of the D3U framework is to decouple the deterministic and uncertain components in time series data without imposing specific methods for obtaining these components. In our paper, only the two case studies in Section 1 and the experiment in Section B.2 of the Appendix use the series decomposition technology (referred to as DCP) to acquire the trend/seasonal components and residual components of the time series.
>
> The use of DCP in Section 1's case studies assists our analysis of the point forecasting model's ability to capture different components of time series data with varying levels of uncertainty. These two case studies indicate that point forecasting models have better modeling capability for components with higher determinacy than those with higher uncertainty. This observation inspires us to leverage the complementary strengths of point forecasting and diffusion models in time series data modeling. The experiment on two hyperparameters (kernel $k$ and Fourier factor $f$) of DCP in Appendix B.2 aims to demonstrate how different hyperparameter settings affect the final performance when using the high-certainty component $x_T + x_S$ ($x_T$ and $x_S$ represent the trend and seasonal components, respectively) as the input for point forecasting models. The optimal parameter settings vary across different point forecasting models. This experiment provides foundational guidance for practitioners considering using $x_T + x_S$ as input within the D3U framework. Additionally, given that some point forecasting models, such as DLinear $\underline{[1]}$, inherently integrate DCP in their design, we retain the option to use $x_T + x_S$ as model input.
>
> In the rest of the paper, we do not use DCP to model high-certainty or high-uncertainty components of time series data. As shown in the experimental results of Table 9 (Section B.2), the effectiveness of DCP is highly dependent on the choice of hyperparameters ($k$ and $f$). Therefore, we do not consider DCP a reliable approach for extracting high-certainty or high-uncertainty components in all cases.
>
> We have opted for a more straightforward and more effective approach: using the entire data $x$ as input for the point forecasting model and treating the model's prediction error as the high-uncertainty component, which is then used as input for the generative model. As analyzed in Section 2.1, the prediction error of the point forecasting model primarily consists of two types of uncertainty: aleatoric uncertainty and epistemic uncertainty. Thus, the prediction error is a reasonable approximation of the high-uncertainty component in the time series data that is difficult for the point forecasting model to capture.
>
> [1] Zeng A, Chen M, Zhang L, et al. Are transformers effective for time series forecasting?[C]//Proceedings of the AAAI conference on artificial intelligence. 2023, 37(9): 11121-11128.

---

> ### Author Response · Authors · 2024-11-22
>
> Dear Reviewer jQCt,
>
> We appreciate it if you could let us know whether our responses are able to address your concerns. We're happy to address any further concerns. Thank you,
>
> Best wishes.
>
> Paper7497 Authors

---

> ### Comment · Reviewer_jQCt · 2024-11-25
> **Response**
>
> Dear Authors,
> thank you for the elaborate reply which answers my questions.

---

> > ### Author Response · Authors · 2024-11-25
> >
> > Dear Reviewer jQCt,
> >
> > We are pleased to have addressed your concerns. In light of the changes made, could you kindly reconsider your score? Thank you for your time and consideration.
> >
> > Paper7497 Authors

---

> > ### Author Response · Authors · 2024-11-26
> >
> > Dear Reviewer jQCt,
> >
> > We’ve noticed that the discussion period for the conference has been extended. If you have any further concerns or suggestions, please don’t hesitate to share them with us. We’re more than happy to address them to the best of our ability. Thank you for your valuable time and consideration!
> >
> > Best wishes.
> >
> > Paper7497 Authors

---

### Official Review · Reviewer_PGSo · 2024-11-04

**Soundness:** 3
**Presentation:** 3
**Contribution:** 2
**Rating:** 6
**Confidence:** 3

**Summary:**

This work introduces a novel diffusion-based time series framework called D3U, designed to enhance probabilistic forecasting. D3U effectively decouples the deterministic and uncertain components of the forecasting process. The conditional network is focused on delivering precise point forecasts, while the conditional DDPM (PatchDN) handles high-uncertainty probabilistic forecasting. Additionally, D3U functions as a plug-and-play framework, allowing for easy integration into existing point forecasting models. Experimental results demonstrate substantial improvements over SOTA baselines.

**Strengths:**

1. The motivation behind this work is strong. The approach of decoupling point and probabilistic forecasting is well-founded.
2. The design of the two components to address different forecasting targets is logical and effective.
3. The experiments are comprehensive, and the improvements in performance are notable.
4. The writing and presentation are clear and easy to follow.

**Weaknesses:**

1. A more detailed discussion of the differences between this work and related studies is essential to clarify its contributions. For example, TMDM also integrates deterministic transformer and diffusion models.
2. Additionally, the selection of baselines could be more comprehensive. Models such as TimeGrad and CSDI, which are relevant probabilistic forecasting approaches, are not included. Currently, the focus is primarily on non-autoregressive models.

**Questions:**

Given that other works also integrate diffusion models with deterministic models, what distinguishes your approach from theirs? How is your model superior in design? For example, how does it compare to models like TAMA, which you mentioned in your paper? And what about CARD[1] ?

[1]Han, Xizewen, Huangjie Zheng, and Mingyuan Zhou. "Card: Classification and regression diffusion models." Advances in Neural Information Processing Systems 35 (2022): 18100-18115.

---

> ### Author Response · Authors · 2024-11-19
> **Question**
>
> Thank you for your valuable comments and suggestions.
> The primary distinction between the D3U framework and other approaches using point forecasting models lies in the decoupling of deterministic and uncertain components in time series data. Based on the two case studies in Section 1, we observed that point forecasting models are more adept at capturing components with lower uncertainty while struggling with components characterized by higher uncertainty. Inspired by this, the D3U framework adopts an uncertainty decoupling approach, leveraging the complementary strengths of point forecasting and diffusion models. Specifically, D3U uses point forecasting models to capture components with lower uncertainty and employs probabilistic forecasting models to handle components with higher uncertainty. The prediction error of the point forecasting model is considered as the high-uncertainty components that the model fails to capture, and a conditional denoising diffusion probabilistic model (DDPM) is then used to model the distribution of these components. The final prediction is generated by combining the outputs of the point forecasting model and the conditional diffusion model. *To the best of our knowledge, D3U is the first framework to employ this uncertainty decoupling approach for long-term multivariate time series (MTS) probabilistic forecasting.*
>
> One of the key strengths of the D3U framework is its use of a simple uncertainty decoupling modeling approach that leads to significant performance improvement. By applying the D3U framework, the corresponding point forecasting model retains its original point forecasting capability and achieves superior probabilistic forecasting performance. In our paper, we conducted experiments to compare the performance of the D3U framework with the TMDM framework on long-term MTS forecasting tasks. The results in Table 3 show that, under identical model configurations (using NSformer as the point forecasting model and MLP as the denoising network for the diffusion model), the D3U framework outperforms TMDM in both point forecasting and probabilistic forecasting capabilities.

---

> ### Author Response · Authors · 2024-11-19
> **Question**
>
> In CARD, a pre-trained conditional mean estimator is injected into both the forward and reverse diffusion chains to construct a DDPM. Specifically, the endpoint of the diffusion process is assumed to be $p(y^k|x)=\mathcal{N}(f_\phi(x),\mathbf{I})$, where $x$ denotes the input and  $y^0$ denotes the ground-truth of the output, $f_\phi (x)$ is the prior knowledge of the relation between $x$ and $y^0$. The forward process conditional distributions are specified as:
> \begin{equation}
> q(y^k│y^{k-1},f_ϕ (x) )=\mathcal{N}(y^k;\sqrt{1-β_k} y^{k-1}+(1-\sqrt{1-β_k }) f_ϕ (x),β_k \mathbf{I}).
> \end{equation}
> The backward process conditional distributions are specified as:
> \begin{equation}
> q(y^{k-1}│y^k,y^0,f_ϕ (x) )=\mathcal{N}(y^{k-1};\tilde{μ}(y^k,y^0,f_ϕ (x)),\tilde{β}_k \mathbf{I}).
> \end{equation}
>
> TMDM adopts a design similar to CARD and modifies the starting point of the reverse process in the standard diffusion model from $\mathcal{N}(0, \mathbf{I})$ to $\mathcal{N}(\hat{y_{1:L}} , \mathbf{I})$ is the output of the point forecasting model. The forward process of TMDM is specified as:
> \begin{equation}
> q(y_{1:L}^k│y_{1:L}^{k-1},\hat{y_{1:L}})=\mathcal{N}(y_{1:L}^k;\sqrt{1-\beta_k}y_{1:L}^{k-1}+(1-\sqrt{1-β_k })\hat{y_{1:L}},\beta_k\mathbf{I}).
> \end{equation}
> The backward process of TMDM is specified as:
> \begin{equation}
> q(y_{1:L}^{k-1}│y_{1:L}^k,y_{1:L}^0, \hat{y_{1:L}})=\mathcal{N}(y_{1:L}^{k-1};\tilde\mu(y_{1:L}^k,y_{1:L}^0, \hat{y_{1:L}}), \tilde{β}_k \mathbf{I}),
> \end{equation}
>
> where $\tilde\mu(y_{1:L}^k,y_{1:L}^0, \hat{y_{1:L}})=\dfrac{\beta_k\sqrt{\bar\alpha_{k-1}}}{1-\bar\alpha_k}y_{1:L}^0+
> \dfrac{\sqrt\alpha_k(1-\bar\alpha_{k-1})}{1\bar\alpha_k}y_{1:L}^k+\left(1+\dfrac{(\sqrt{\bar\alpha_k}-1)(\sqrt\alpha_k+
> \sqrt{\bar\alpha_{k-1}})}{1-\bar\alpha_k}\right)\hat{y_{1:L}}$,
> and $\bar\alpha_k:=\prod_{i=1}^{k}\alpha_k$ with $\alpha_i:=1-\beta_i$.
>
> In TMDM, $\hat{y_{1:L}}$ serves as prior knowledge, providing a good initial estimate for the reverse process, which helps accelerate the denoising of the data. Essentially, TMDM uses DDPM to model the distribution of the entire dataset, rather than decoupling and separately modeling the uncertainty components as done in D3U.
>
> D3U employs the original conditional DDPM, where the starting point of the reverse process is $\mathcal{N}(0, \mathbf{I})$.  Instead of modeling $y^0$ directly, DDPM in D3U models $r^0 := y^0-\hat{y_{0:L}}$, where $r^0$ represents the residual between the true value and the point forecast $\hat{y_{0:L}}$. The forward process of D3U is specified as:
> \begin{equation}
> q(r_{1:L}^k│r_{1:L}^{k-1})=\mathcal{N}(r_{1:L}^{k-1};\sqrt{1-\beta_k}{r_{1:L}^{k-1}}, {β}_k\mathbf{I}).
> \end{equation}
>
> D3U injects the time series representation $f_{enc}(x)$ obtained from the pre-trained point forecasting model into the denoising network of the reverse process. The backward process of D3U is specified as:
> \begin{equation}
> q(r_{1:L}^{k-1}│r_{1:L}^k,f_{enc}(x))=\mathcal{N}(r_{1:L}^{k-1};\tilde\mu(r_{1:L}^k,k|f_{enc}(x)),\tilde\beta_k \mathbf{I}).
> \end{equation}
>
> where $\tilde\mu(r_{1:L}^k,k|f_{enc}(x))=\dfrac{\sqrt\alpha_k(1-\bar\alpha_{k-1})}{1-\bar\alpha_k}r_{1:L}^k+\dfrac{\sqrt{\bar\alpha_{k-1}}\beta_k}{1-\bar\alpha_k}r_\phi(r^k_{1:L},k|f_{enc}(x))$ and $f_{enc}(x)$ is given by the denoising network.

---

> ### Author Response · Authors · 2024-11-19
> **Weakness 2**
>
> Thank you for your valuable suggestions. In light of your comment, we add two highly-cited baselines, TimeGrad $\underline{[1]}$ and CSDI $\underline{[2]}$, to our paper.  The experimental results, now integrated into the updated version, are presented as follows ("**Bold**/*italic* denotes the **best**/*second-best* results."):
>
> Table 1: Performance comparison on six real-world datasets based on MSE and MAE in probabilistic forecasting models.
> |     Dataset    |      ETTm1      |      ETTm2      |     Weather     |   Solar-Energy  |   Electricity   |     Traffic     |
> |:--------------:|:---------------:|:---------------:|:---------------:|:---------------:|:---------------:|:---------------:|
> |     Method     |     MSE MAE     |     MSE MAE     |     MSE MAE     |     MSE MAE     |     MSE MAE     |     MSE MAE     |
> | TimeGrad(2021) |   1.716 1.057   | 1.385 0.732     | 0.885 0.551     | 1.211 1.004     | 0.645 0.723     | 0.932 0.807     |
> |   CSDI(2021)   | 0.867 0.690     | 1.291 0.576     | 0.842 0.523     | 0.848 0.818     |  0.553 0.795    | 0.921 0.678     |
> | TimeDiff(2023) |   0.796 0.577   |  _0.284 0.342_  |   0.277 0.331   |   1.169 0.936   |   0.730 0.690   |   1.465 0.851   |
> |   TMDM(2024)   |  _0.607 0.558_  |   0.524 0.493   |  _0.244 0.286_  |  _0.295 0.317_  |  _0.222 0.329_  |  _0.721 0.411_  |
> |      ours      | **0.363 0.386** | **0.241 0.302** | **0.222 0.264** | **0.237 0.270** | **0.179 0.267** | **0.468 0.299** |
>
> Table 2: Performance comparisons on six real-world datasets regarding CRPS and CRPS_sum in probabilistic forecasting models.
> |     Dataset    |      ETTm1      |      ETTm2      |     Weather     |   Solar-Energy  |    Electricity    |      Traffic      |
> |:--------------:|:---------------:|:---------------:|:---------------:|:---------------:|:-----------------:|:-----------------:|
> |     Method     |  CRPS CRPS_sum  |  CRPS CRPS_sum  |  CRPS CRPS_sum  |  CRPS CRPS_sum  |   CRPS CRPS_sum   |   CRPS CRPS_sum   |
> | TimeGrad(2021) |   0.665 0.996   |   0.785 1.051   |   0.482 0.503   |   0.783 1.167   |    0.503 1.452    |    0.657 1.683    |
> |   CSDI(2021)   |   0.773 0.852   |   0.625 0.782   |   0.508 0.465   |   0.649 0.681   |    0.465 0.823    |    0.612 1.275    |
> | TimeDiff(2023) |   0.454 0.846   |  _0.316 0.180_  |   0.293 0.400   |   0.900 1.164   |    0.475 0.594    |    0.671 0.823    |
> |   TMDM(2024)   |  _0.429 0.633_  |   0.380 0.226   |  _0.226 0.292_  |  _0.375 0.267_  | _0.446_ **0.137** | _0.552_ **0.179** |
> |      ours      | **0.285 0.574** | **0.243 0.141** | **0.207 0.141** | **0.186 0.266** | **0.202** _0.160_ | **0.232** _0.186_ |
>
>
> The results shown in the table above demonstrate that, compared to the other four models, D3U exhibits superior point prediction accuracy and distribution estimation capability. TimeGrad $\underline{[1]}$  and CSDI $\underline{[2]}$ are widely utilized in the field of short-term time series forecasting. When applied to long-term time series forecasting, the masking-based conditional generative model CSDI outperforms the autoregressive model TimeGrad, improving point prediction accuracy and probabilistic forecasting capability by 16% and 17%, respectively. However, it is noteworthy that the performance of D3U surpasses both TimeGrad and CSDI by a significant margin. The reasons for this significant outperformance by D3U are detailed as follows:
>
> 1. D3U decouples the deterministic and uncertain components in multivariate time series (MTS) data, leveraging the complementary strengths of point forecasting models and diffusion models to model these two parts separately. Within the D3U framework, the diffusion model focuses on capturing the high-uncertainty components of the data, while the information related to the more deterministic components is provided by the point forecasting model. This design not only preserves the point forecasting ability of D3U but also reduces the learning difficulty for the diffusion model.
>
> 2. The proposed PatchDN enhances the diffusion model's ability to capture both the guidance information from the point forecasting model and the temporal features of time series data.
>
> [1] Rasul K, Seward C, Schuster I, et al. Autoregressive denoising diffusion models for multivariate probabilistic time series forecasting[C]//International Conference on Machine Learning. PMLR, 2021: 8857-8868.
>
> [2] Tashiro Y, Song J, Song Y, et al. Csdi: Conditional score-based diffusion models for probabilistic time series imputation[J]. Advances in Neural Information Processing Systems, 2021, 34: 24804-24816.

---

> ### Author Response · Authors · 2024-11-22
>
> Dear Reviewer PGSo,
>
> We appreciate it if you could let us know whether our responses are able to address your concerns. We're happy to address any further concerns. Thank you,
>
> Best wishes.
>
> Paper7497 Authors

---

> > ### Comment · Reviewer_PGSo · 2024-11-25
> > **Response**
> >
> > I appreciate the authors for their efforts in addressing my concerns. Most of the issues I raised have been resolved. So I have decided to raise my score accordingly.

---

> > > ### Author Response · Authors · 2024-11-25
> > >
> > > Dear Reviewer PGSo,
> > >
> > > We sincerely thank you for your careful and detailed reading of our paper and responses.
> > >
> > > Best regards,
> > >
> > > Paper7497 Authors

---

### Meta-Review · Area_Chair_kGN4 · 2024-12-21

**Metareview:**

This paper introduces the D$^3$U framework for probabilistic multivariate time series (MTS) forecasting. D$^3$U employs a decoupling strategy to independently model deterministic and uncertain components in time series data. Specifically, it integrates a point forecasting model to address high-certainty components with a conditional diffusion model (DDPM) to capture high-uncertainty components. Experimental results on six real-world datasets show that this approach enhances both point and probabilistic forecasting, outperforming several state-of-the-art baselines.

**Pros**:
The D$^3$U framework demonstrates competitive or superior performance across diverse datasets, advancing state-of-the-art results in long-term probabilistic forecasting. Its modular, plug-and-play design ensures flexibility, allowing seamless integration with existing forecasting models and workflows.

**Cons**:
Despite its strong empirical performance, the underlying methodology bears significant resemblance to Transformer-Modulated Diffusion Models (TMDM, Li et al., ICLR 2024). Notably, D$^3$U also leverages a pretrained model for point estimates and employs diffusion models for uncertainty modeling, similar to TMDM. To strengthen its contribution, the paper should explicitly distinguish D$^3$U from TMDM, especially in explaining the theoretical or practical reasons for its superior performance. This differentiation needs to be integrated into the motivation section of the paper, rather than being deferred to the experimental results as is currently done.

Overall, the experimental results are robust, with D$^3$U consistently outperforming state-of-the-art baselines across multiple datasets. The authors were diligent in addressing reviewer concerns, providing additional baseline comparisons and detailed visualizations to clarify the respective roles of deterministic and uncertain components. While the theoretical and methodological innovation is incremental, the framework represents a meaningful advancement in the practical application of diffusion models for probabilistic forecasting, making it a valuable contribution to the field.

**Additional Comments On Reviewer Discussion:**

The rebuttal period brought attention to the following key discussion points:

- **Comparison with TMDM, CARD, and Related Works**: Several reviewers sought clarification on the theoretical distinctions and improvements over TMDM. The authors provided detailed explanations, emphasizing the decoupling of uncertainty components and the residual modeling approach. From the AC's perspective, the paper needs to clearly articulate the theoretical distinction between initiating the reverse process from a point estimate (as in TMDM) versus starting from zero and modeling the residual in reverse diffusion (as in D$^3$U). Without this clarification, the observed performance gains could potentially be attributed to confounding factors not adequately addressed in the rebuttal.

- **Effectiveness of the Diffusion Model**: Concerns were raised about whether the diffusion model genuinely enhances predictions or merely introduces uncertainty. The authors provided visualizations and probabilistic metrics (e.g., CRPS) to support their claims, though some reviewers remained skeptical. The AC does not find this point concerning and is satisfied with how the authors addressed this point.

- **Baseline Comparisons**: Reviewers noted the absence of TimeGrad and CSDI in the initial comparisons. The authors responded by including these baselines, demonstrating that D$^3$U consistently outperforms them, further substantiating its effectiveness.


The authors' thorough responses and additional experiments effectively addressed most of the concerns, resulting in a positive shift in reviewer sentiment. However, the AC remains somewhat skeptical regarding how D$^3$U fundamentally differs from TMDM. Despite this, the AC is inclined to support the paper due to its convincing performance gains and the robustness of its experimental results.

---

### Decision · Program_Chairs · 2025-01-22

Accept (Poster)